# COCO-GAN: CONDITIONAL COORDINATE GENERATIVE ADVERSARIAL NETWORK

## ABSTRACT

Recent advancements on Generative Adversarial Network (GAN) have inspired a wide range of works that generate synthetic images. However, current processes have to generate an entire image at once, and therefore resolutions are limited by memory or computational constraints. In this work, we propose COnditional COordinate GAN (COCO-GAN), which generates a specific patch of an image conditioned on a spatial position rather than the entire image at a time. The generated patches are later combined together to form a globally coherent full-image. With this process, we show that the generated image can achieve competitive quality to state-of-the-arts and the generated patches are locally smooth between consecutive neighbors. One direct implication of the COCO-GAN is that it can be applied onto any coordinate systems including the cylindrical systems which makes it feasible for generating panorama images. The fact that the patch generation process is independent to each other inspires a wide range of new applications: firstly, "Patch-Inspired Image Generation" enables us to generate the entire image based on a single patch. Secondly, "Partial-Scene Generation" allows us to generate images within a customized target region. Finally, thanks to COCO-GAN's patch generation and massive parallelism, which enables combining patches for generating a full-image with higher resolution than state-of-the-arts.

## 1 INTRODUCTION

This paper explores the idea of enforcing both the generator and the discriminator of generative adversarial networks (GANs) (Goodfellow et al., 2014) to deal with only partial views via conditional coordinating. Via training and inference with partial views only, the minimum memory requirement can be largely reduced. However, as shown in Section 3.3, naive approaches fail to generate high quality images, either having clear seams or totally failing to generate reasonable structures. We investigate this problem and propose a new GAN architecture: COnditional COordinate GAN (COCO-GAN).

Given a latent vector and multiple spatial positions, the generator of COCO-GAN learns to produce image patches independently according to the spatial positions. On the other hand, the discriminator learns to judge whether adjacently generated patches are structurally sound and visually homogeneous. During the inference phase, the generated patches can directly be used to compose a complete full-image without further post-processing. Owing to the adversarial loss provided by the discriminator, the composed full-image is locally smooth and globally convincing. We show several randomly-selected full-images generated by COCO-GAN in Figure 2a. In Section 3.2, we visualize the interpolation between two spatial positions, showing COCO-GAN has inter-class continuity as common conditional GANs do. Further quantitative evaluations with "Frchet Inception Distance" (FID) (Heusel et al., 2017) score are presented in Table 1. Without additional hyper-parameter tuning, the evaluations on CelebA (Liu et al., 2015) and LSUN (Yu et al., 2015) datasets suggest that COCO-GAN is competitive with other state-of-the-art GANs. To further demonstrate the effectiveness of COCO-GAN, we perform ablation study in Section 3.3.

In Section 3.4, we demonstrate COCO-GAN is a flexible coordinate-system-aware framework that is suitable for different spatial coordinate systems. Common learning frameworks are confined to

---

The code and data used in the paper will be made available immediately upon publication.

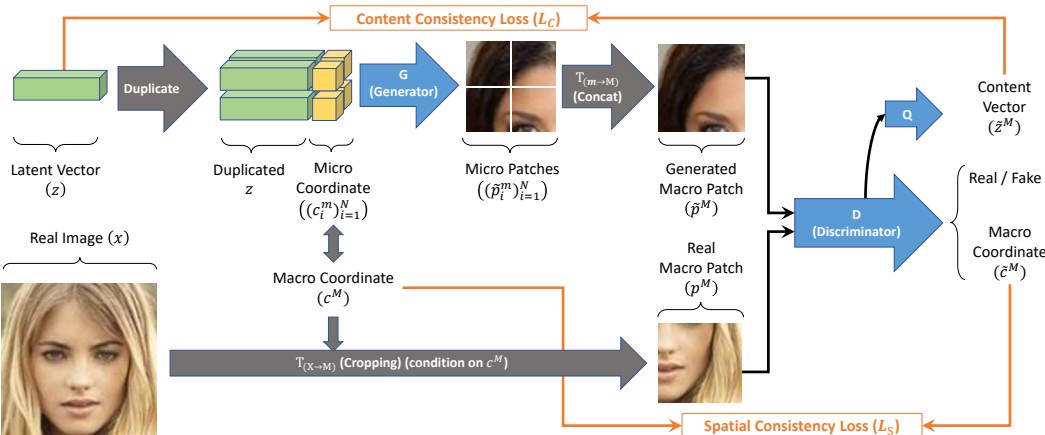

Figure 1: An overview of COCO-GAN (training phase). The full-images are only generated during testing phase (Figure 9 in the Appendix).

Cartesian coordinate system, which restricts those frameworks from learning the characteristics of certain image formats. For instance, panoramas should be trained with a cylindrical coordinate system, which has a "cyclic topology" in the horizontal direction. In comparison, COCO-GAN directly learns with a cylindrical coordinate system. In Figure 6 and Figure 14, the generated full scenes are naturally cylindrical and continuous while crossing the left and right borders.

Besides, we find COCO-GAN has multiple interesting applications and characteristics. We select three representative new applications as case studies:

**Patch-Inspired Image Generation** takes a real image patch as input. COCO-GAN can be inspired by the given patch and further generates a full-image proposal. This proposal is partially similar to the given patch, while globally realistic and reasonable. This setting is different from conventional image completion/reconstruction since the information loss during cropping is extreme and the spatial position information is not provided to the model. Therefore, COCO-GAN has to infer the position of the given patch before generating the whole image. Further experiments are presented in Section 3.5.

**Partial-Scene Generation** shows that COCO-GAN can generate partial scenes without spending additional computation outside certain designated regions. This capability is exclusively beneficial to applications that are only interested in partial information of the full scene. For instance, virtual reality (VR) is only interested in the user's viewport direction, and COCO-GAN can seamlessly adapt to such viewport-aware settings.

**Computation-Friendly Generation** demonstrates the computational related merits. First, since the patches are produced independently, the generator is able to generate patches with high parallelism. Second, as the full-image generation is decomposed to patches generation, the minimum memory requirement of generating images can be reduced. This characteristic enables COCO-GAN to generate images of very high resolution or containing much more complex structures.

## 2 COCO-GAN

**Overview.** COCO-GAN consists of three networks: a generator $G$, a discriminator $D$, and a auxiliary head $Q$. These three networks are trained with four loss terms: patch Wasserstein loss $L_W$, patch gradient penalty loss $L_{GP}$, spatial consistency loss $L_S$, and content consistency loss $L_C$. Compared to conventional GANs that use full images as input for both $G$ and $D$, COCO-GAN only uses micro patches for $G$, and macro patches for $D$. The details of micro and macro patches will be described in the paragraph "Spatial coordinate system design". $D$ has two auxiliary prediction heads: the content vector prediction head ($Q$) and the spatial condition prediction head. $Q$ is trained

with an extra optimizer independent to $G$ and $D$. It aims to minimize $L_C$ and is used to estimate the original latent vector of the given sample. The spatial condition prediction head, on the other hand, is jointly trained with the discriminator. It aims at minimizing $L_S$ and is used to estimate the macro spatial position of the given sample. Both of the two auxiliary prediction heads are simple feed-forward networks that take inputs from one of the high-level feature maps of the discriminator.

COCO-GAN considers two coordinate systems: a micro coordinate system $m$, which refers to the annotated component that related to the finer coordinate system, on the generator's side, and a macro coordinate system $M$, which is related to the coarser macro coordinate system on the discriminator's side. The discriminator learns to distinguish between the generated macro patch $\tilde{p}^M$ and the real macro patch $p^M$. The discriminator also learns to predict two auxiliary outputs: a predicted macro coordinate $\tilde{c}^M$ and a predicted latent vector $\tilde{z}^M$. Outputs from these two networks are then used to compute two auxiliary losses: Spatial Consistency Loss ($L_S$) and Content Consistency Loss ($L_C$).

The objective function of the discriminator $D$ is $L_W + L_{GP} + L_S + L_C$, the generator $G$ is $-L_W + L_S + L_C$, and the content vector prediction head $Q$ is $-L_W + L_C$.

**Spatial coordinate system.** Before presenting the details of these four loss terms, we introduce our notations first. We start with designing two spatial coordinate systems, a *micro* coordinate system $C^m$ for the generator $G$ and a *macro* coordinate system $C^M$ for the discriminator $D$. Let $S^N$ be a space of spatial position sequences, each spatial position sequence $s = \langle c_i^m \rangle_{i=1}^N \in S^N$ is an ordered sequence, which $c_i^m \in C^m$. During COCO-GAN training, $R$ is some predefined spatial constraints for sampling $s$ from a uniform distribution $\mathcal{U}(S^N, R)$. The generator $G$ is conditioned by each spatial position $c_i^m$, and learns to accordingly produce micro patches $\tilde{p}_i^m = G(z|c_i^m)$. The $\langle \tilde{p}_i^m \rangle_{i=1}^N = \langle G(z|c_i^m) \rangle_{i=1}^N$ is a sequence of micro patches produced ***independently*** while sharing the same latent vector $z$ across the spatial position sequence $s$.

The design of $R$ may need to be slightly changed with respect to the selection of $C^M$ and $C^m$. The design principle of $R$ is that the accordingly generated micro patches $\langle \tilde{p}_i^m \rangle_{i=1}^N$ should be spatially close to each other. Then the micro patches are merged by a merging function $T_{(m \to M)}$ to form a complete macro patch $\tilde{p}^M = T_{(m \to M)}(\langle \tilde{p}_i^m \rangle_{i=1}^N)$ as a coarser partial-view of the full-scene $\tilde{x}$. Meanwhile, we assign $\tilde{p}^M$ with a new spatial position $c^M$ under the macro coordinate system for $s$.

In Figure 1, we illustrate one of the simplest design for the above heuristic functions that we have adopted throughout our experiments. The four micro patches are always a neighbor of each other and can be directly combined into a square macro patch with $T_{(m \to M)}$, which is just simple concatenation. Figure 3 shows some examples of micro and macro patches generation.

On the real samples side, we also sample $s \sim S$, but we directly infer $s$ to macro position $c^M$. Then we design a transformation $T_{(X \to M)}$ with respect to $T_{(m \to M)}$ to transform a real full-image $x$ to a real macro patch $p^M$ by $T_{(X \to M)}(x|c^M)$. The shape of $p^M$ should be the same as $\tilde{p}^M$. In our simplest experiment setting, $T_{(X \to M)}$ is a simple cropping function that crops $x$ into $p^M$, which has the same shape as $\tilde{p}^M$.

During the testing phase, depending on the design of $C^m$, we can directly infer a corresponding spatial position sequence $\langle c_j^m \rangle_{j=1}^K$. It is used to independently produce spatially-disentangled patches that constitute the full-image. Figure 9 demonstrates how the full-image is generated during the testing phase. Figure 2a shows some examples of the full-image generation.

**Loss functions.** The patch Wasserstein loss $L_W$ is a patch-level Wasserstein distance loss similar to Wasserstein-GAN (Arjovsky et al., 2017) loss. It forces the discriminator to discriminate between the real macro patch $p^M$ and the generated macro patch $\tilde{p}^M$, and on the other hand, encourages the generator to confuse the discriminator with seemingly realistic $\tilde{p}^M$. Its complete form is

$$L_W = \mathbb{E}\left[ D(T_{(X \to M)}(x|c^M)) \right] - \mathbb{E}\left[ D(T_{(m \to M)}(\langle G(z|c_i^m) \rangle_{i=1}^N)) \right], \tag{1}$$

where $x \sim \mathbb{P}_r$ and $\langle c_i^m \rangle_{i=1}^N \sim \mathcal{U}(S^N, R)$. Note that $\mathbb{P}_r$ is the real data distribution. We also apply Gradient Penalty (Gulrajani et al., 2017) to the patches generation:

$$L_{GP} = \mathbb{E}\left[ (\|\nabla_{\tilde{p}^M} D(\tilde{p}^M)\|_2 - 1)^2 \right], \tag{2}$$

where $\tilde{p}^M \sim \mathbb{P}_g$. Note that $\mathbb{P}_g$ is the generator distribution.

The spatial consistency loss $L_S$ is similar to ACGAN loss (Odena et al., 2017). A slight difference is that $c_i^m$ has relatively more continuous values than the discrete setting of ACGAN. As a result, we apply a distance measurement loss for $L_S$, which is an $L_2$-loss between $\tilde{c}^M$ and $c^M$. It aims to train the generator of COCO-GAN to generate the corresponding micro patch by $G(z|c_i^m)$ with respect to the given spatial condition $c_i^m$. The spatial consistency loss is

$$L_S = \quad \mathbb{E}\left[\|c^M - \tilde{c}^M\|_2\right] . \tag{3}$$

On the other hand, the content consistency loss $L_C$ is similar to a hybrid of infoGAN loss (Chen et al., 2016) and the latent space constraint loss (Chang et al., 2018). The former one uses a separate optimizer to optimize an auxiliary network $Q$, which aims to reconstruct the original latent vector. The latter one suggests that the latent space consistency loss can be a distance measurement instead of minimizing the KL-divergence as the original infoGAN does. In our experiments, we train the extra $Q$ network with a separate optimizer and directly minimize $L_1$-loss between $\tilde{z}$ and $z$. $L_C$ aims to force the generator to produce shared context between patches that share the same latent vector but locate at different micro coordinate positions. The content consistency loss is defined by

$$L_C = \quad \mathbb{E}\left[\|z - \tilde{z}\|_1\right] . \tag{4}$$

To ensure that $Z$, $C^m$, and $C^M$ share the similar scale, which are directly concatenated and feed to $G$. We evaluate the maximum possible pixel position of $C^m$ and $C^M$, then normalize the range into $[-1, 1]$. For the latent space $Z$, although uniform sampling between $[-1, 1]$ should be numerically more compatible with the normalized spatial condition space, we empirically do not observe significant differences even if we switch to random sampling with a zero-mean and unit-variance normal distribution. For simplicity, we adopt uniform sampling strategy throughout our experiments.

**Training details.** Our generator and discriminator architectures follow the idea of projection discriminator (Miyato & Koyama, 2018), both with ResNet (He et al., 2016) based architecture and adding class-projection to the discriminator. All convolutional and feed-forward layers of generator and discriminator are added with the spectral-normalization scheme (Miyato et al., 2018) as suggested in (Zhang et al., 2018). A more detailed architecture diagram is illustrated in Appendix B.

We also add conditional-batch-normalization (CBN) (Dumoulin et al., 2016) to the generator. In our design, CBN is conditioned on the given spatial positions and the input latent vector. It learns to normalize the feature maps with respect to the given conditions. However, our implementation has a crucial difference from the one described in (Miyato & Koyama, 2018): our spatial positions are real values rather than discrete classes. We alternatively adopt similar strategy to (de Vries et al., 2017) with a slight modification. Instead of using MLPs to produce $\Delta\gamma$ and $\Delta\beta$, we make MLPs directly output $\gamma$ and $\beta$. For a $K$-channel input feature map $i_K$ with mean recorder $\mu_K$ and variance recorder $\sigma_K$, we creates two learnable MLP layers, $\mathrm{MLP}_\gamma$ and $\mathrm{MLP}_\beta$. The output feature map $o_M$ is calculated as $o_M = ((i_K - \mu_K)/\sigma_K) * \mathrm{MLP}_\gamma(i_K) + \mathrm{MLP}_\beta(i_K)$.

We use Adam (Kingma & Ba, 2014) optimizer with $\beta_1 = 0$ and $\beta_2 = 0.999$ for both the generator and the discriminator. The learning rates are based on the Two Time-scale Update Rule (TTUR) (Heusel et al., 2017), setting 0.0001 for the generator and 0.0004 for the discriminator. We do not specifically balance the generator and the discriminator by manually setting how many iterations to update the generator once as described in the WGAN paper (Arjovsky et al., 2017).

## 3 EXPERIMENTS

Although COCO-GAN framework supports the spatial positions to be uniformly and continuously sampled within normalized range $[-1, 1]$, we empirically find that only sampling the discrete spatial positions that is used to inference will result in better generation quality. For instance, if the full-image is formed by concatenating four micro patches on each axis, the model only uniformly samples spatial positions from the set of four discrete points, $\{-1, -1/3, 1/3, 1\}$ on each axis of the coordinate system. We adopt this uniform and discrete sampling strategy throughout the experiments. The root cause of degradation in generation quality is still unclear. One possible hypothesis is that the task difficulty dramatically increases while sampling with continuous spatial positions. We flag further analysis and solution toward this phenomenon as an important future research direction. Some comparisons between continuous sampling and discrete sampling are shown in Appendix F.

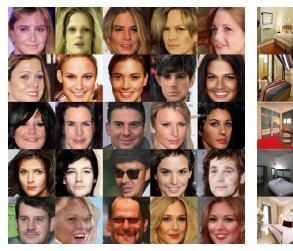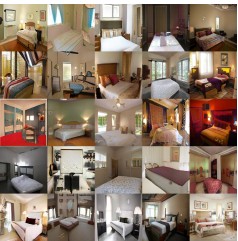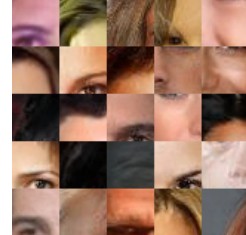

(a) CelebA (128×128).   (b) LSUN (256×256).     (a) Micro patch (32×32). (b) Macro patch (64×64).

Figure 2: Without any post-processing, the generated full-images of COCO-GAN are visually smooth and globally coherent. More full-image generation results are shown in Figure 12.

Figure 3: Generated samples of micro/macro patches. Each micro patch, macro patch, and full-image in Figure 2a at the same relative position uses the same latent vector.

Table 1: The FID score suggests that COCO-GAN is competitive with other state-of-the-art GANs. FID scores are measured between 50,000 real and generated samples based on the original implementation provided at https://github.com/bioinf-jku/TTUR.

| Dataset | DCGAN + TTUR | PGGAN | WGAN-GP + TTUR | COCO-GAN |
|---|---|---|---|---|
| CelebA (64×64) | 12.5 | - | - | **4.99** |
| CelebA (128×128) | - | **7.30** | - | 8.35 |
| LSUN - Bedroom (64×64) | 57.5 | - | **9.5** | - |
| LSUN - Bedroom (128×128) | - | - | - | **3.06** |
| LSUN - Bedroom (256×256) | - | **8.34** | - | 16.59 |

### 3.1 QUALITY OF GENERATED IMAGES

We start with validating COCO-GAN on CelebA (Liu et al., 2015) and the bedroom category of LSUN (Yu et al., 2015). For CelebA dataset, the resolutions of full-image, micro patch, and macro patch are $128 \times 128$, $32 \times 32$, and $64 \times 64$, respectively. We choose $32 \times 32$ for micro patches in this experiment since smaller patch size would be too small for the model (neither for human) to observe useful information. On the other hand, larger patch size makes macro patch size too similar to the full-image size, which is hard to demonstrate the idea of COCO-GAN can learn without access to the full-image. For LSUN dataset, the full-image is with $256 \times 256$ resolution, micro patches with $64 \times 64$ resolution and macro patches with $128 \times 128$ resolution. We choose $64 \times 64$ for micro patch size since the micro patch to full-image ratio is the same with the CelebA experiment.

We report Frchet Inception Distance (FID) (Heusel et al., 2017) in Table 1 as quantitative comparisons with state-of-the-art GANs. Since many other state-of-the-art models do not use full-resolution of the datasets, we accordingly run COCO-GAN in different resolutions without changing hyperparameters other than input size and micro/macro patch size. Throughout these experiments, we choose to always retain the micro patch size to be 1/16 (1/4 for height and 1/4 for width) of the full-image size and macro patch size to be 1/4 (1/2 for height and 1/2 for width) of the full-image size. Without additional hyper-parameter tuning, the results suggest COCO-GAN is both qualitatively and quantitatively competitive with other state-of-the-art GANs.

In Appendix H, we also provide Wasserstein distance and FID score through time as training indicators. The curves suggest that COCO-GAN is stable during training.

### 3.2 LATENT SPACE CONTINUITY

To demonstrate more precisely the space continuity, we perform the interpolation experiment in three directions: micro patches interpolation, spatial positions interpolation, and full-images interpolation.

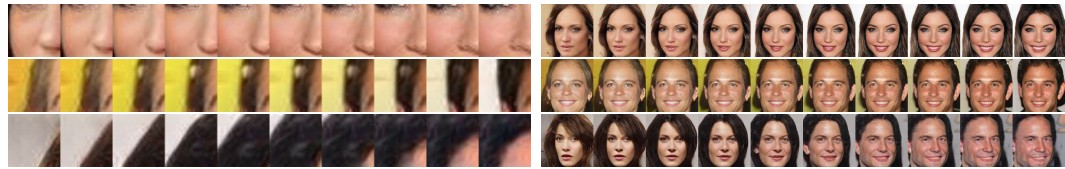

| (a) Micro patches interpolation. | (b) Full images interpolation. |

Figure 5: (a) Micro patches interpolation with fixed spatial position. Note that each left-right pair of interpolation sample uses the same latent vectors. (b) Full-images interpolation between two latent vectors. More interpolation results are shown in Appendix D.

**Micro Patches Interpolation.** The simplest interpolation experiment is the in-class (*e.g.* fixed spatial condition) interpolation between latent vectors. With a fixed spatial position $c_i^{micro}$, we randomly sample two latent vectors $z_1$ and $z_2$. Then perform interpolation between $z_1$ and $z_2$ through a slerp-path (White, 2016). The results in Figure 5a suggest that for each of the spatial position, the latent space $Z$ has continuity.

**Spatial Positions Interpolation.** Another simple interpolation experiment is inter-class (*e.g.* between classes) interpolation with a fixed latent vector. We directly linearly-interpolate spatial position between $[-1, 1]$ when the latent vector $z$ is fixed. The results in Figure 4 show that, although we only uniformly sample spatial positions within a discrete spatial position set, the spatial position interpolation is still continuous.

An interesting observation is about the interpolation at the position between the eyebrows. In this example, COCO-GAN does not know the existence of the smooth area (glabella) between two eyes due to the discrete and sparse spatial positions sampling strategy. Instead, it learns to directly deform the shape of eye to switch from one eye to another. This phenomenon raises an interesting discussion, even the model learns to produce high-quality face images, it still may learn wrong relationship of objects behind the scene.

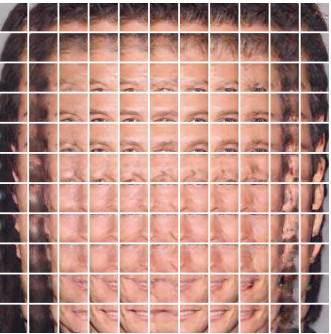
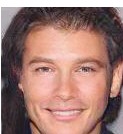

Figure 4: An example of spatial positions interpolation for showing the spatial continuity of the micro patches. The spatial conditions are interpolated between range $[-1, 1]$ of the micro coordinate with a fixed latent vector. More examples are shown in Appendix I.

**Full-Images Interpolation.** The hardest interpolation is to directly interpolate full-images between two latent vectors. All micro patches generated with different spatial positions must all change synchronously to make the full-image interpolation smooth. We randomly sample two latent vectors $z_1$ and $z_2$. With any given interpolation point $z'$ in the slerp path between $z_1$ and $z_2$, the generator uses the full spatial position sequence $\langle c_j^m \rangle_{j=1}^K$ to generate all corresponding patches. Then we merge all generated micro patches with $T_{(m \to M)}$ and forms a full-image $x'$. The interpolation results in Figure 5a and Figure 5b show that all micro patches can interpolate smoothly and synchronously. This result suggests that COCO-GAN learns the main latent space $Z$ as well as the correlation between micro patches, and the spatial conditions $C^m$ are disentangled.

## 3.3 ABLATION STUDY

The ablation study is conducted in two folds: we first show that a straightforward approach fails in COCO-GAN setting, then we study the trade-offs of each component of COCO-GAN.

**A straightforward approach.** One straightforward approach to the learning and inference with partial views is using a full-sized generator and a patch discriminator, and then pre-calculating the

locations of feature maps that are associated with a specific partial view. We refer to this method as $\mathcal{M}$ afterward. Despite $\mathcal{M}$ still implicitly uses conditional coordinating in feature map selection, we observe that it fails to generate high-quality samples in comparison with COCO-GAN. We show the FID score results in Table 2. More experimental details are shown in Figure 15 and some generated samples in Figure 16.

**The trade-offs of each component.** We perform ablation study in CelebA $64 \times 64$ setting with five configurations: "continuous sampling" demonstrates that using continuous uniform sampling strategy for spatial positions during training will result in moderate generation quality drop; "optimal $D$" lets the discriminator directly discriminate the full image while the generator still generates micro patches; "optimal $G$" lets the generator directly generate the full image while the discriminator still discriminates macro patches; "without $Q$" removes the $Q$ network from COCO-GAN; "multiple $G$" trains an individual generator for each spatial position.

Table 2: Best FID scores in the first 150 epochs. COCO-GAN usually converges well in CelebA $64 \times 64$ setting.

| Model | FID |
|---|---|
| $\mathcal{M}$ | 72.82 |
| $\mathcal{M}$ + PD (100 epochs) | 90.87 |
| $\mathcal{M}$ + PD + macro $D$ | 60.36 |
| COCO-GAN (cont. sampling) | 6.13 |
| COCO-GAN + optimal $D$ | 4.05 |
| COCO-GAN + optimal $G$ | 6.12 |
| COCO-GAN + without $Q$ | 4.87 |
| Multiple $G$ | 7.26 |
| COCO-GAN (ours) | 4.99 |

The results in Table 2 suggest $Q$ network is not a necessary component if not considering the "Patch-Inspired Image Generation" application. Surprisingly, despite the convergence speed is different, "optimal discriminator", COCO-GAN, and "optimal generator" (ordered by convergence speed from fast to slow) can all achieve similar FID scores if with sufficient training time. The difference in convergence speed is expected, since "optimal discriminator" provides the generator with more accurate and global adversarial loss. In contrast, the "optimal generator" has relatively more parameters and layers to optimize, which causes the convergence speed slower than COCO-GAN. Lastly, the "multiple generators" setting cannot converge well. Although it can also concatenate micro patches without obvious seams as COCO-GAN does, the full-image results often cannot agree and are not globally coherent. More experimental details and generated samples are shown in Figure 17 and Figure 18.

## 3.4 PANORAMA GENERATION AND PARTIAL SCENE GENERATION

Generating panoramas using GANs is an interesting problem but has never been carefully investigated. Different from simple image generation, panoramas are expected to be cylindrical and cyclic in the horizontal direction. However, normal GANs do not have built-in ability to handle such cyclic characteristic if without special types of padding mechanism support (Cheng et al., 2018). In contrast, COCO-GAN is a coordinate-system-aware learning framework. We can easily adapt a cylindrical coordinate system, and generate panoramas that are cyclic in the horizontal direction as shown in Figure 6 and Figure 14.

To train COCO-GAN with a panorama dataset under a cylindrical coordinate system, the spatial position sampling strategy needs to be slightly modified. In the horizontal direction, the sampled value within the normalized range $[-1, 1]$ is treated as an angular value $\theta$, and then is projected with $\cos(\theta)$ and $\sin(\theta)$ individually to form a unit-circle on a 2D surface. Along with the normal sampling on the vertical axis, a cylindrical coordinate system is formed.

We first take the sky-box format of Matterport3D (Chang et al., 2017) dataset to obtain panoramas for training and testing. The sky-boxes consist of six faces of a 3D cube. We preprocess and project the sky-box to a cylinder using Mercator projection, the resulting cylindrical image size is $768 \times 512$. Since the Mercator projection creates extreme sparsity near the northern and southern poles, which lacks information, we directly remove the upper and lower $1/4$ areas. Eventually, the size of panorama we used for training is $768 \times 256$ pixels.

We also find COCO-GAN has interesting connection with virtual reality (VR). VR is known to have tight computational budget due to high frame-rate requirement and high resolution demand. It is hard to generate full-scene for VR in real time using standard generative models. Some recent VR studies on omnidirectional view rendering and streaming (Corbillon et al., 2017b; Ozcinar et al.,

0° 360° 720°

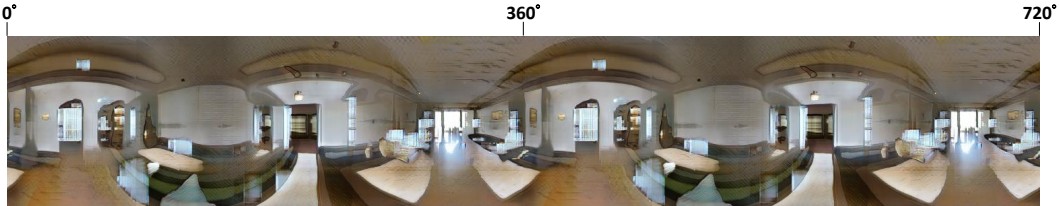

Figure 6: The generated panorama is cyclic in the horizontal direction since COCO-GAN is trained with a cylindrical coordinate system in this experiment. Here we paste the same generated panorama twice (from $360°$ to $720°$) to illustrate that it indeed has the cyclic property.

2017; Corbillon et al., 2017a) are focusing on reducing computational cost or network bandwidth by adapting the user's viewport. COCO-GAN can easily inherit the same strategy and achieve user-viewport-aware partial-scene generation based on its effectiveness in spatial disentanglement and panorama generation. This can largely reduce unnecessary computational cost outside the region of interest, thus making image generation in VR more applicable.

### 3.5 Patch-Inspired Image Generation

The content consistency loss equips the discriminator with the ability to approximate the original latent vector of generated macro patch $\tilde{p}^{macro}$ using the discriminator's auxiliary content vector prediction $\tilde{z}$. This property can also generalize to any real macro patch $p^{macro}$. The approximation process does not require a spatial position; the spatial position is implicitly inferred by the spatial position prediction $\tilde{c}^M$. The generator can accordingly generate a full-image $x'$ with respect to $\tilde{z}$. The generated $x'$ should be partially similar to the given macro patch and also be globally coherent. Such $x'$ can be seen as a generated sample inspired by the given macro patch. One important footnote is that most of the information of a real full-image is lost while retrieving $p^M$. As a result, the produced **guess** of full-image is not guaranteed to be identical to the original real image. We provides some examples of patch-inspired image generation in Figure 7. The results show that $x'$ can loosely retain some local structure or global characteristic of the original image, such as gender, face direction, and facial expression.

This process is also similar to image inpainting (Liu et al., 2018a; Yeh et al., 2017; Yang et al., 2017) except two key differences. First, the spatial position of the macro patch is not explicitly given. Existing image inpainting frameworks assume that the remaining parts of the image are already at their optimal positions, whereas COCO-GAN can infer by itself. Take human face for instance, if only given a cropped patch of a face image without further providing the position of the patch in the original image, COCO-GAN can still infer the position of the patch and reconstruct a full face image, while common inpainting frameworks may not reconstruct a correctly structured and well centered human face. Second, most inpainting frameworks do not assume the image is extremely damaged, like loosing $75\%$ of information in our examples. In Figure 8, we accordingly compare COCO-GAN with *partial convolution* (Liu et al., 2018a), which is one of the state-of-the-art image inpainting methods. For the partial convolution method, we simply place the macro patch at the center since the spatial position of the macro patch is unknown. The results show that, unlike COCO-GAN , the partial convolution method (Liu et al., 2018a) cannot handle this situation well.

### 3.6 Computation-Friendly Generation

Recent studies in high-resolution image generation (Karras et al., 2017; Mescheder et al., 2018) have gained lots of success. We observe a shared conundrum of these existing works is the memory requirement. They usually require some workarounds to improve memory consumption during training, such as decreasing the batch size (Karras et al., 2017) or cutting down the number of feature maps (Mescheder et al., 2018). The memory requirement problem cannot be easily resolved without specific hardware support, which makes the generation of *over* $1024 \times 1024$ resolution images hard to achieve. These types of high-resolution images are commonly seen in panoramas, street views, and medical images.

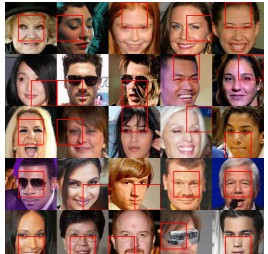 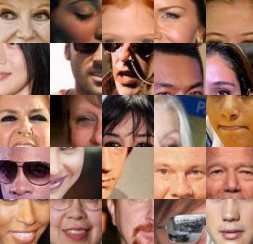 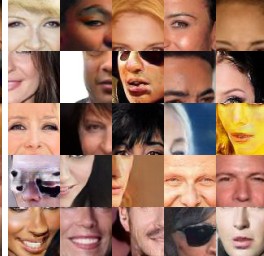 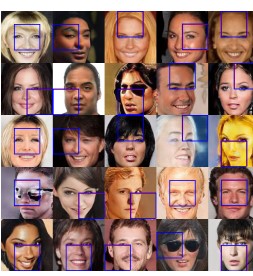

(a) Real full-images.  (b) Real macro patches.  (c) Patch-inspired macro patch generation.  (d) Patch-inspired full-image generation.

Figure 7: Patch-inspired image generation can loosely retain some local structure or global characteristic of the original image. The red boxes are the sampled spatial positions that crop the full-images in (a) into the macro patches in (b). (c) & (d) show the patch-inspired generated macro patches and full-images based on $\tilde{z}$. The blue boxes visualize the predicted spatial positions $\tilde{c}^m$. Since the information loss of the cropping process (from (a) to (b)) is critical, we do not expect (a) and (d) to be identical. Instead, (b) and (c) should be visually similar and (d) should be globally coherent. More examples are shown in Appendix G.

In contrast, COCO-GAN only requires partial views of the full-image for both training and inference. This characteristic largely reduces the minimum memory requirement while dealing with high-resolution images. In Section 3.1, we show that COCO-GAN can generate and compose a $128 \times 128$ image with patches of $32 \times 32$ resolution. In other words, we can train a CIFAR-10-sized model to generate $128 \times 128$ resolution images while the results are still high quality. The characteristic of spatial disentanglement is the first step toward super-high-resolution image generation with limited memory resource. Moreover, some state-of-the-art structures of deep models tend to require more parameters (if without reducing number of channels), such as the skip-connection structure (He et al., 2016) used in projection discriminator (Miyato & Koyama, 2018), inception (Szegedy et al., 2015), and self-attention (Zhang et al., 2018). COCO-GAN is able to equip many of these complex structures since our COCO-GAN model is relatively light-weight with a smaller receptive field and a shallower structure.

Furthermore, the spatial disentanglement makes the generation of micro patches independent after the latent vector and the spatial positions are decided. This characteristic enables the generation of micro patches to have high parallelism and take advantage of modern computation architectures.

## 4 RELATED WORK

Generative Adversarial Network (GAN) (Goodfellow et al., 2014) has shown its potential and flexibility to many different tasks. Recent studies on GANs are focusing on generating high-resolution and high-quality synthetic images in different settings. For instance, generating images with $1024 \times 1024$ resolution (Karras et al., 2017; Mescheder et al., 2018), generating images with low-quality synthetic images as condition (Shrivastava et al., 2017), and by applying segmentation map as conditions (Wang et al., 2017). However, these prior work

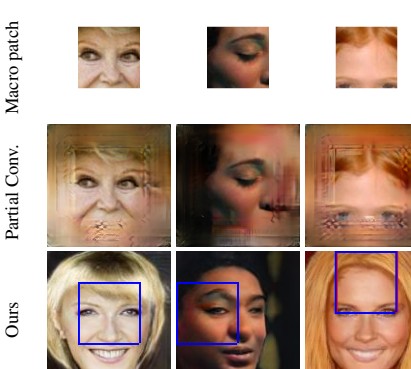

Figure 8: Patch-inspired image generation is globally more coherent than the partial convolution method.

share the similar assumptions: the model must access and generate the full-image in a single shot. This assumption consumes an unavoidable and significant amount of memory when the size of the targeting image is relatively large, and therefore making it difficult to satisfy memory requirements for both training and inference. Searching for a solution towards this problem is one of the initial motivations of this work.

COCO-GAN shares some similarities to Pixel-RNN (van den Oord et al., 2016), which is a pixel-level generation framework while COCO-GAN is a patch-level generation framework. Pixel-RNN transforms the image generation task into a sequence generation task, maximizes the log-likelihood directly. In contrast, COCO-GAN aims at disentangling the spatial dependencies between micro patches. It also utilizes the adversarial loss to ensure smoothness between adjacent micro patches.

CoordConv (Liu et al., 2018b) is another similar work but with fundamental differences. Coord-Conv provides spatial positioning information directly to the convolutional kernels in order to the coordinate transform problem and shows multiple improvements in different tasks. In contrast, COCO-GAN uses spatial conditions as an input condition of the generator and an auxiliary output of the discriminator. This setting enforces both the generator and the discriminator to learn coordinating and correlations between the generated micro patches. We have also considered incorporating CoordConv into COCO-GAN . However, empirical results show little visual improvement.

Group convolution (Krizhevsky et al., 2012) is another work that is highly related to COCO-GAN . While group convolution aims at reducing computational costs by disentangling channels inside a convolution layer, our model learns to disentangle on the spatial level and is highly parallelizable. However, the micro-patch generation of COCO-GAN uses padding in all feature maps while applying convolution. This problem causes a large number number of FLOPs for each image generation. We are particularly interested in this phenomenon and flag utilizing the spatial disentanglement to reduce the total number of FLOPs as an important future work.

# 5    CONCLUSION AND DISCUSSION

In this work, we propose COCO-GAN , a new generative model toward dividing full image generation into non-overlapping patches generation. Through the experiments, we show that COCO-GAN can learn and inference with limited partial views. Although the model is restrained from accessing the full scene, it can still generate high-quality samples without extra hyper-parameter tuning. We also demonstrate COCO-GAN is a coordinate-system-aware framework, and take panorama generation within a cylindrical coordinate system as case study. Furthermore, we highlight the advantages of COCO-GAN by showcasing three applications, including "Patch-Inspired image generation", "User-Viewport-Aware Partial Scene Generation", and "Computation-Friendly Generation".

Despite the generation quality of COCO-GAN being competitive with other state-of-the-art GANs without any post-processing, sometimes we still observe that local structures of generated samples may be discontinued or mottled. This indicates that extra refinements and blending methods are still important for COCO-GAN to generate more stable and reliable samples.

We adopt a discrete uniform sampling strategy over spatial positions since we observe a huge drop in generation quality with continuous uniform sampling. Although in practice COCO-GAN successfully learns spatial continuity using discrete sampling, continuous sampling, in theory, should still be preferred since the spatial domain is continuous. Achieving such a goal would require deeper understanding and insights about the root cause of the generation quality drop.

We demonstrate that COCO-GAN can generate panoramas under a cylindrical coordinate system. However, another commonly used panorama format is sky-sphere under a hyperbolic coordinate system. Considering that the image patches with the hyperbolic coordinate is not square-shaped, further studies on incorporating special convolution schemes like Spherical-CNN (Cohen et al., 2018) and implementing COCO-GAN under a hyperbolic coordinate system would be required. Furthermore, to allow a more flexible and general coordinate system, some learnable coordinating methods (Balakrishnan et al., 2018) might be correlated to COCO-GAN and could further enhance the flexibility.

The size of micro patches is crucial to the results of COCO-GAN. A rule-of-thumb is that the size should be large enough to cover sufficient information. The precise lower bound requires experiments to examine if COCO-GAN learns undesired spatial patterns. In "Spatial Positions Interpolations" of Section 3.2, we mention the model can be misled to learn reasonable but incorrect spatial relationship. An effective evaluation for the lower bound of the patch size needs future investigation.

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

## APPENDIX A   COCO-GAN DURING TESTING PHASE

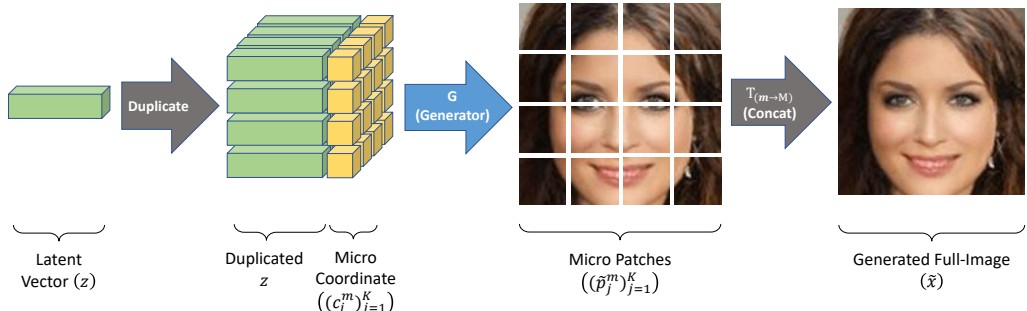

Figure 9: An overview of COCO-GAN during testing phase. The micro patches generated by $G$ are directly combined into a full-image as the final output.

## APPENDIX B   MODEL ARCHITECTURE DETAILS

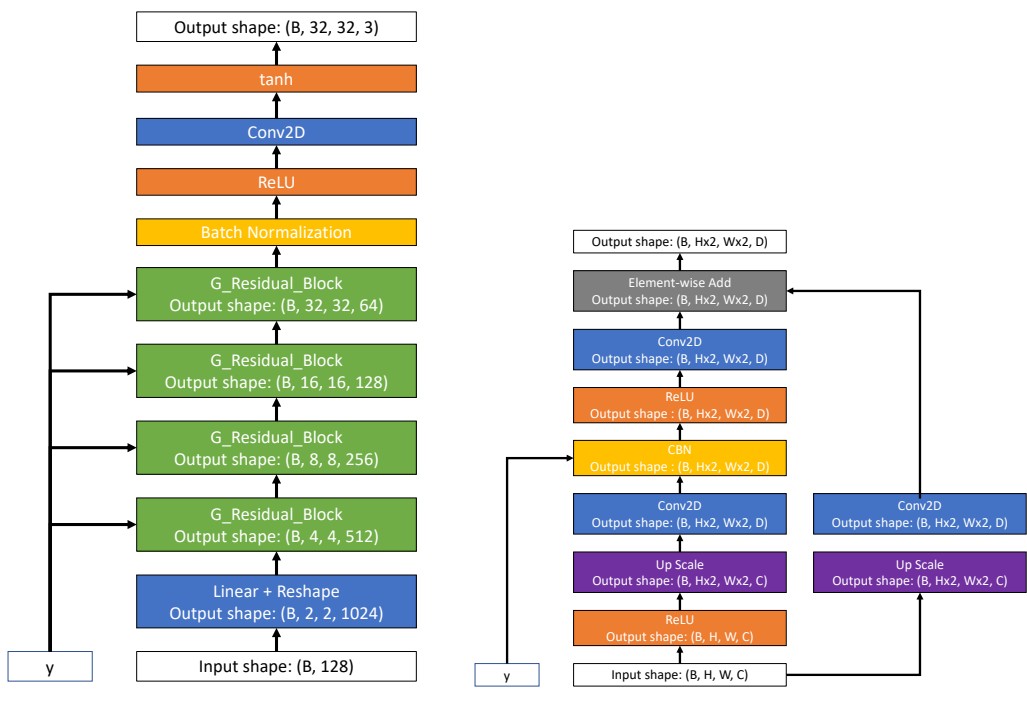

(a) Generator Overall Architecture          (b) Generator Residual Block

Figure 10: The detailed generator architecture of COCO-GAN for generating micro patches with a size of $32 \times 32$ pixels. We directly duplicate/remove the last residual block if we need to enlarge/reduce the size of output patch.

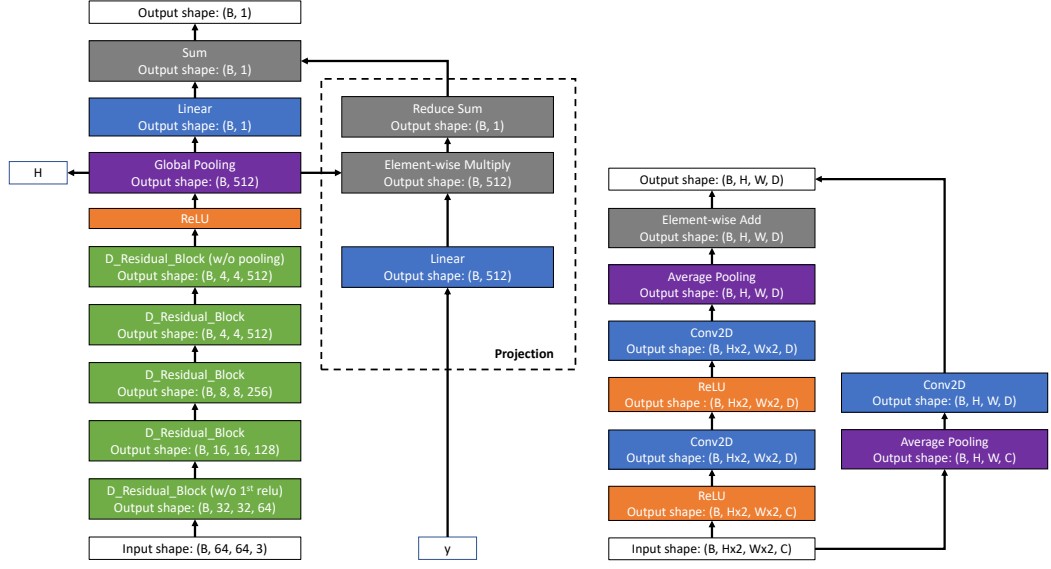

(a) Discriminator Overall Architecture      (b) Discriminator Residual Block

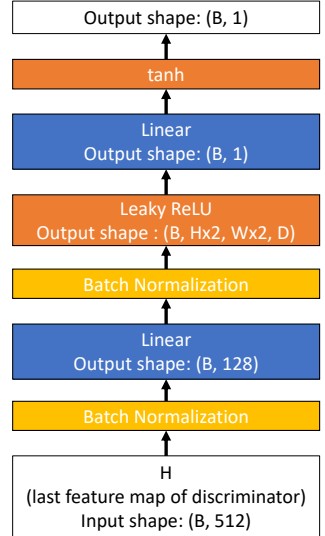

(c) Discriminator Auxiliary Head

Figure 11: The detailed discriminator architecture of COCO-GAN for discriminate macro patches with a size of $64 \times 64$ pixels. We directly duplicate/remove the first residual block if we need to enlarge/reduce the input patch size. Both the content vector prediction head ($Q$) and the spatial condition prediction head use the same structure shown in (c).

APPENDIX C   MORE FULL-IMAGE GENERATION EXAMPLES

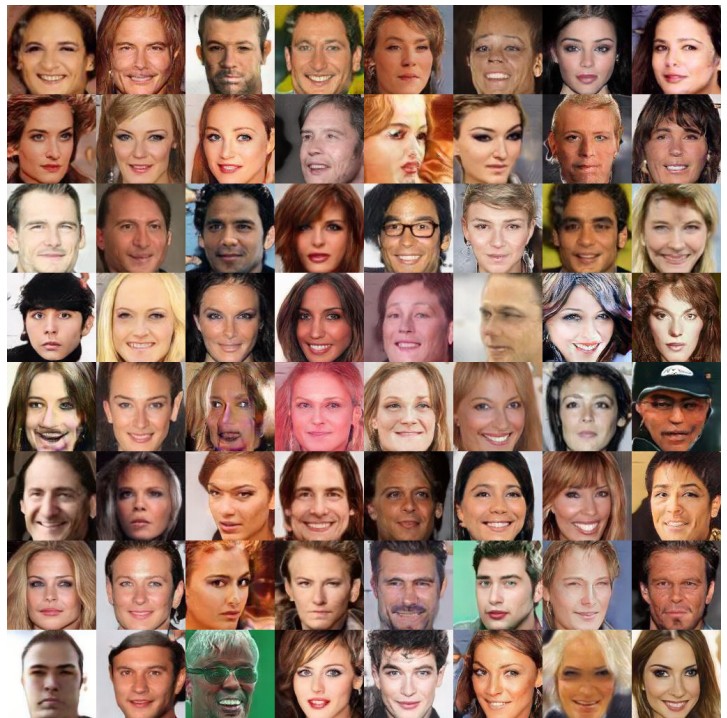

(a) CelebA $128 \times 128$

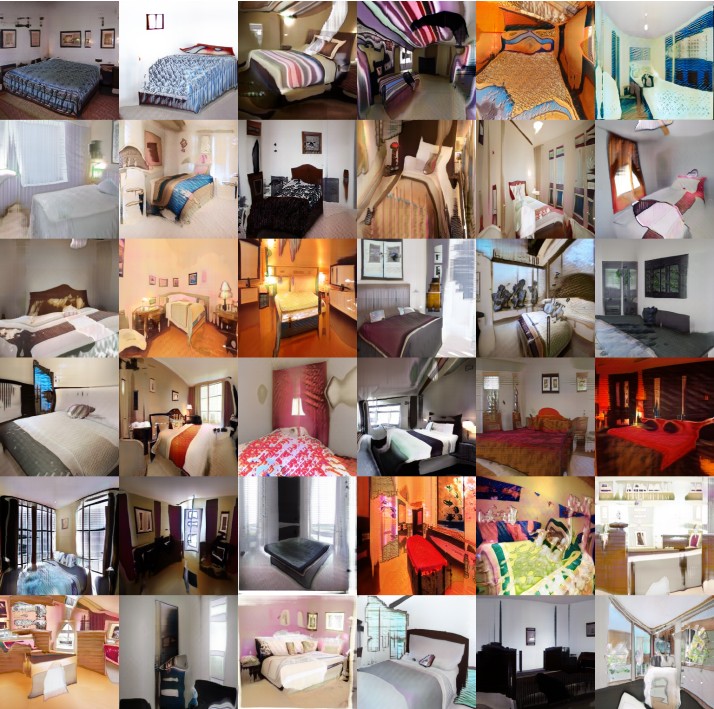

(b) LSUN (bedroom) $256 \times 256$

Figure 12: More full-image generation examples of COCO-GAN. More results across epochs are provided in following anonymous link: https://goo.gl/A88ewn and https://goo.gl/hgCAeE.

## APPENDIX D    MORE INTERPOLATION EXAMPLES

**Micro Patches Interpolation**    **Full-Images Interpolation**

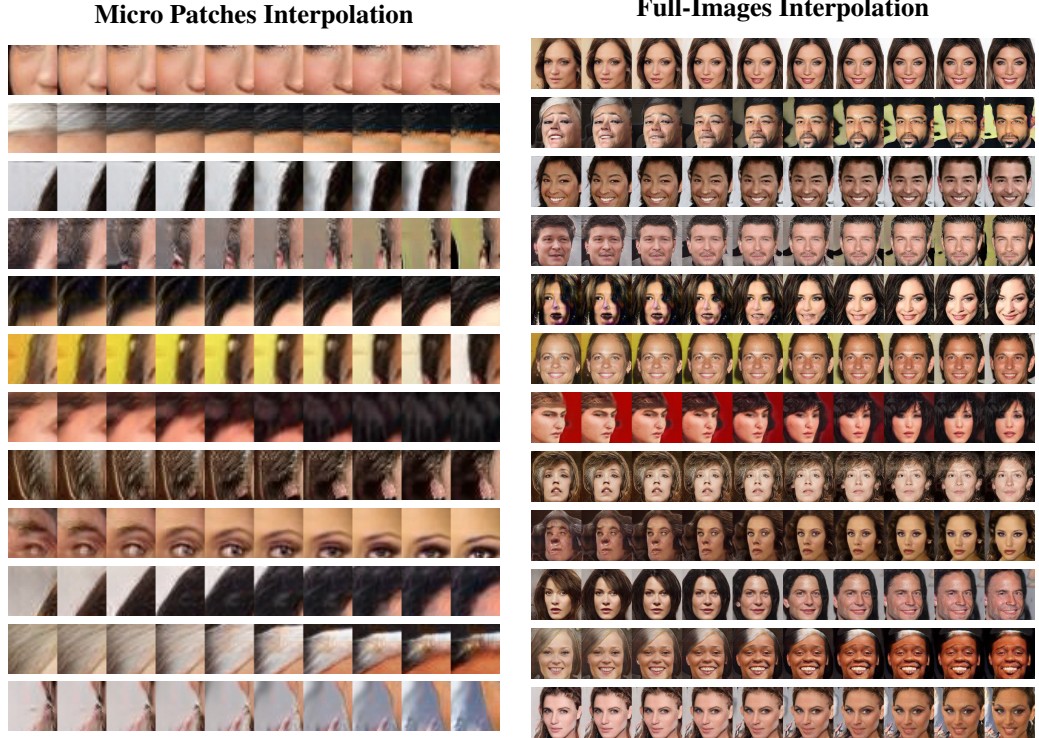

(a) CelebA (128 × 128).

**Micro Patches Interpolation**    **Full-Images Interpolation**

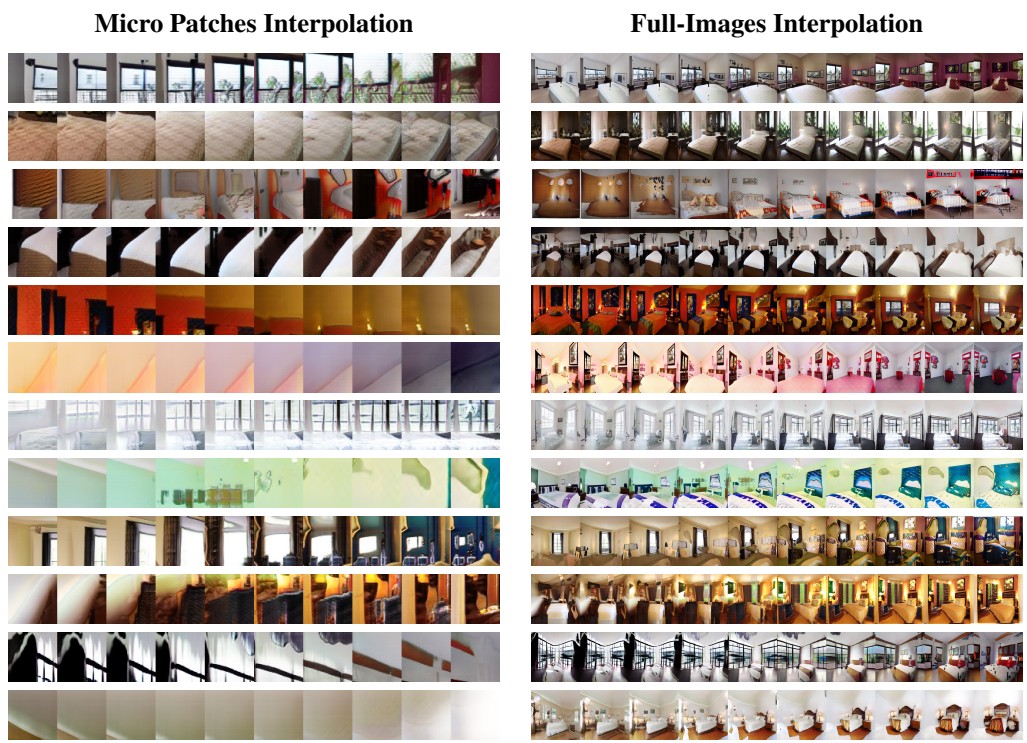

(b) LSUN (bedroom category) (256 × 256).

Figure 13: More interpolation examples. Given two latent vectors, COCO-GAN generates the micro patches and full-images that correspond to the interpolated latent vectors.

## APPENDIX E   MORE PANORAMA GENERATION SAMPLES

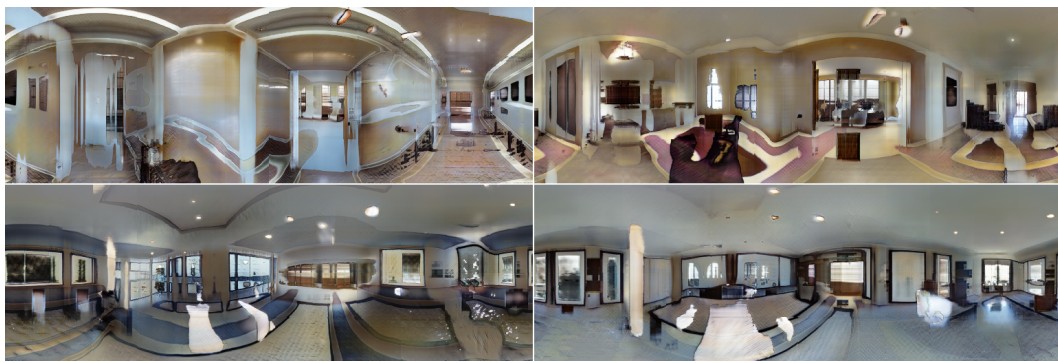

Figure 14: More examples of generated panoramas. All samples possess the cyclic property along the horizontal direction. Each sample is generated with a resolution of $768 \times 256$ pixels, and micro patch size $64 \times 64$ pixels.

## APPENDIX F   ABLATION STUDY

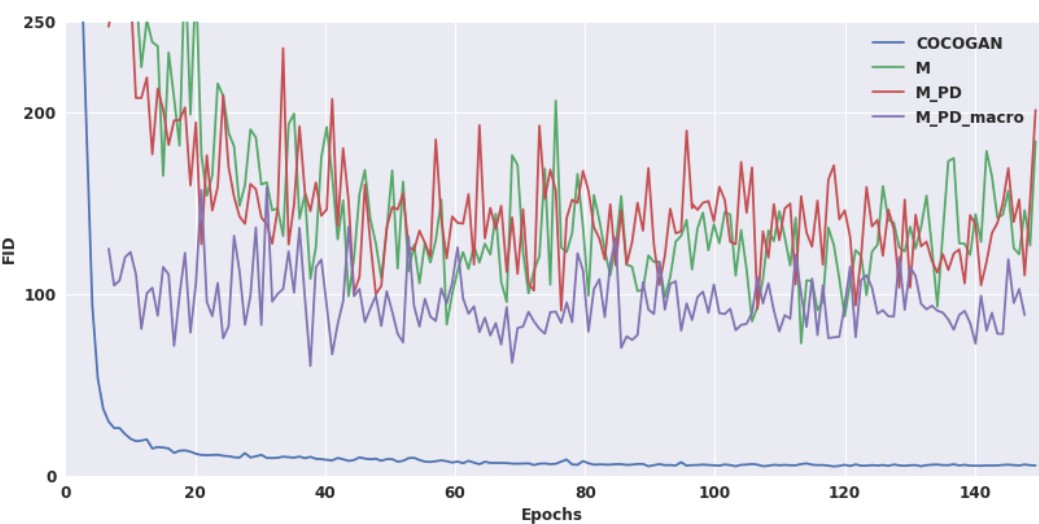

Figure 15: Comparison with the $\mathcal{M}$ method mentioned in Section 3.3 in CelebA $64 \times 64$ setting shows that the $\mathcal{M}$ method is not competitive to COCO-GAN. Note that PD refers to "projection discriminator" and macro indicates the discriminator is in macro patch sized.

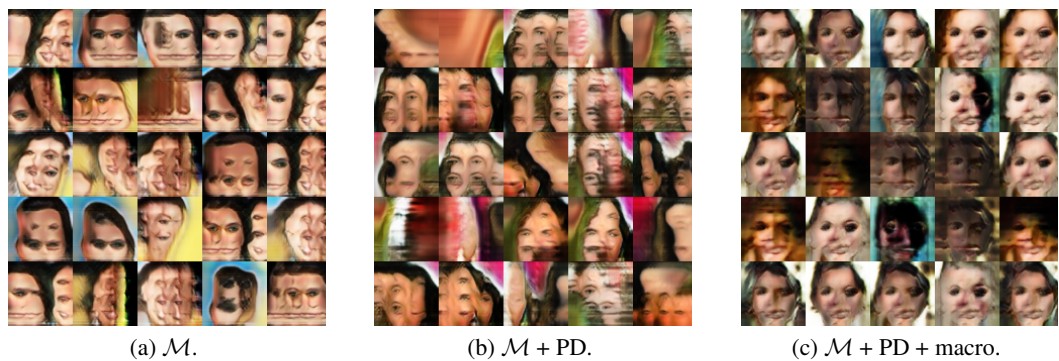

(a) $\mathcal{M}$.     (b) $\mathcal{M}$ + PD.     (c) $\mathcal{M}$ + PD + macro.

Figure 16: Some samples generated by different variants of $\mathcal{M}$. Note that each set of samples is extracted at the epoch when each $\mathcal{M}$ variant reaches its lowest FID score. We also provide more samples at different epochs: https://goo.gl/ChQhCx.

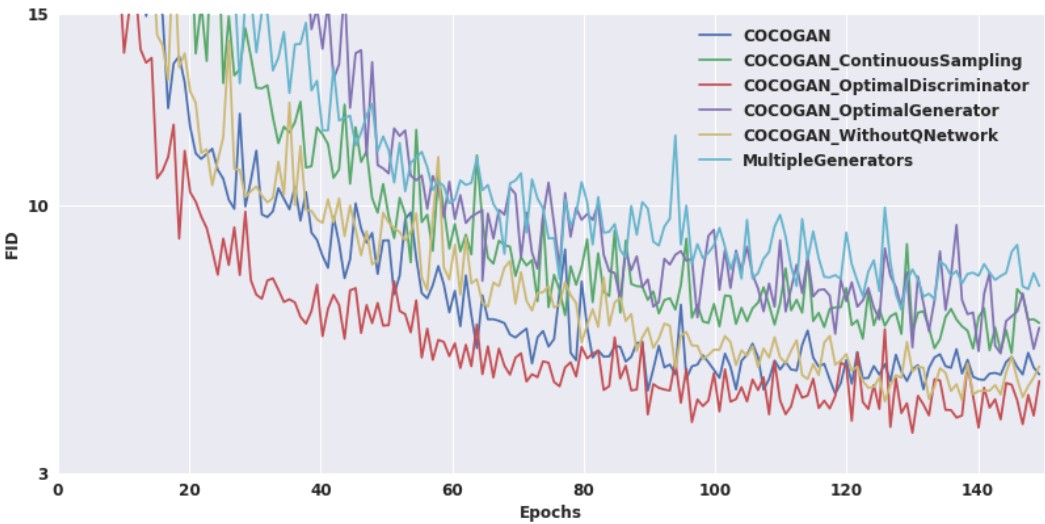

Figure 17: FID score curves of different variants of COCO-GAN in CelebA $64 \times 64$ setting. Combined with Figure 18, the results do not show significant differences in quality between COCO-GAN variants. Therefore, COCO-GAN does not pay significant trade-off for the conditional coordinate property.

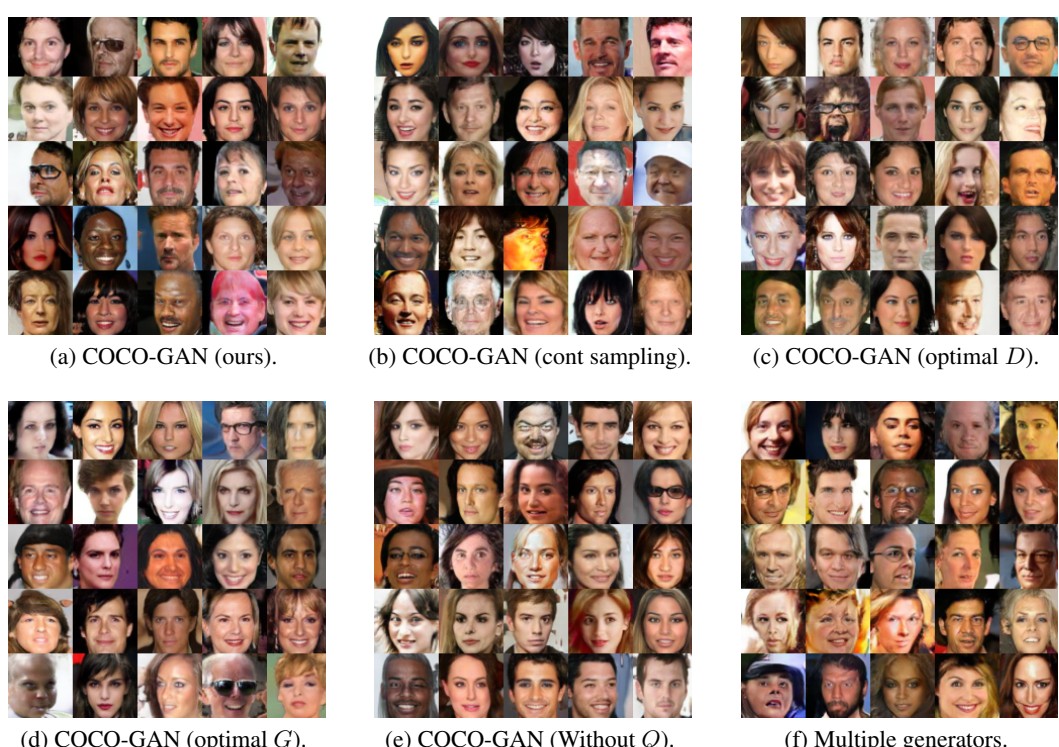

(a) COCO-GAN (ours).     (b) COCO-GAN (cont sampling).     (c) COCO-GAN (optimal $D$).

(d) COCO-GAN (optimal $G$).     (e) COCO-GAN (Without $Q$).     (f) Multiple generators.

Figure 18: Some samples generated by different variants of COCO-GAN. Note that each set of samples is extracted at the epoch when each $\mathcal{M}$ variant reaches its lowest FID score. We also provide more samples for each of the variants at different epochs: https://goo.gl/Wnrppf.

## APPENDIX G   PATCH-INSPIRED IMAGE GENERATION

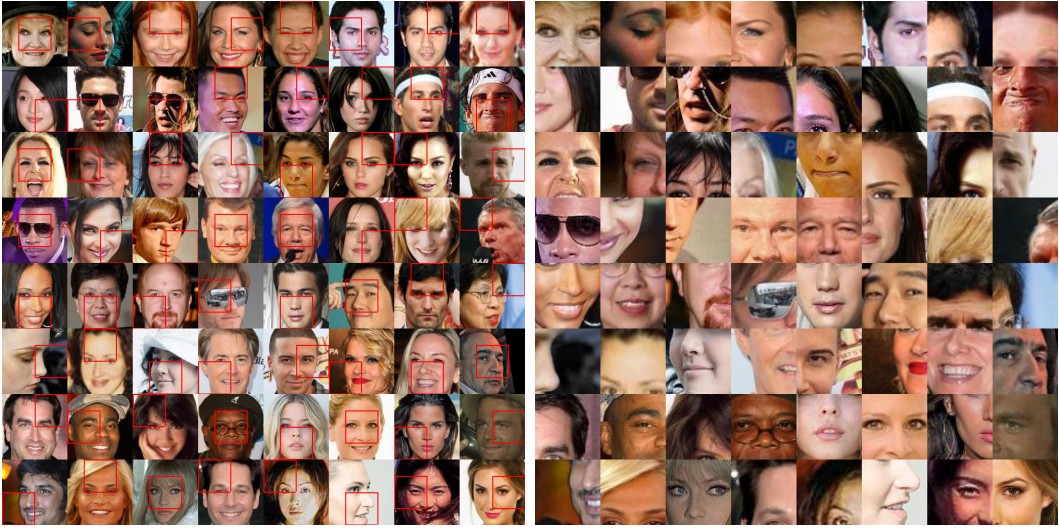

(a) (CelebA 128×128) Real full-images.          (b) (CelebA 128×128) Real macro patches.

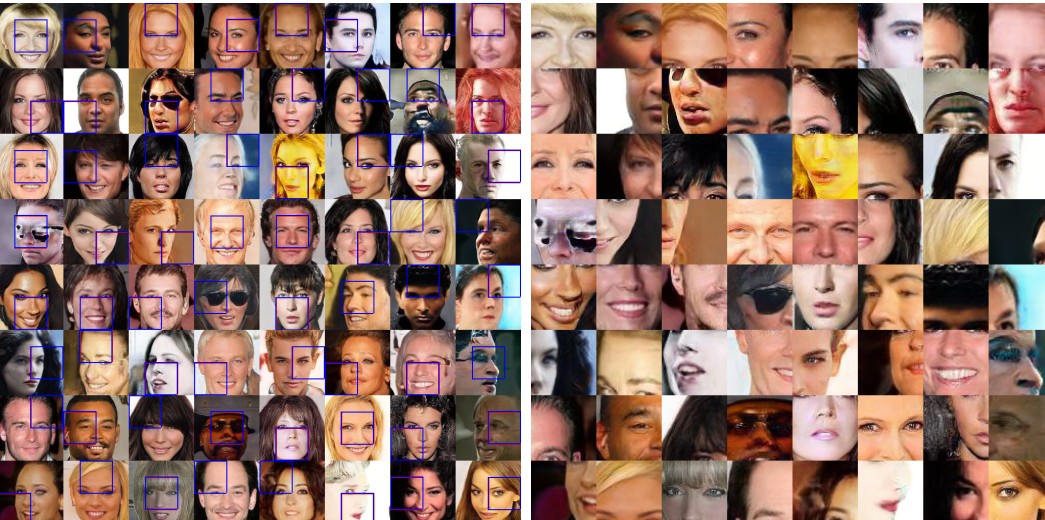

(c) (CelebA 128×128) Patch-inspired full-image generation.          (d) (CelebA 128×128) Patch-inspired macro patch generation.

Figure 19: Patch-inspired image generation can loosely retain some local structure or global characteristic of the original image. The red boxes are the sampled spatial positions that crop the full-images in (a) into the macro patches in (b). (c) & (d) show the patch-inspired generated macro patches and full-images based on $\tilde{z}$. The blue boxes visualize the predicted spatial position $\tilde{c}^m$. Since the information loss of the cropping process (from (a) to (b)) is critical, we do not expect (a) and (c) to be identical. Instead, (b) and (d) should be visually similar and (d) should be globally coherent.

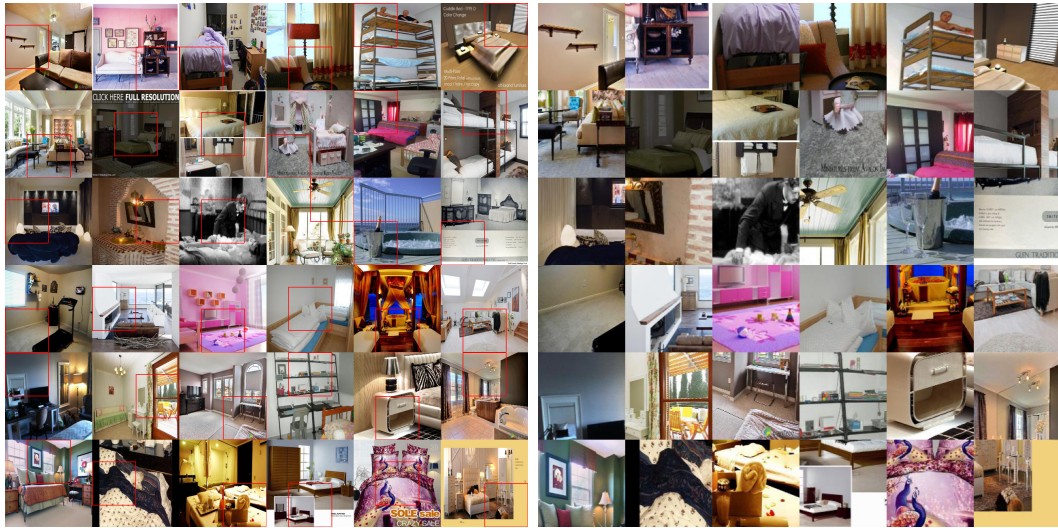

(a) (LSUN 256×256) Real full-images.

(b) (LSUN 256×256) Real macro patches.

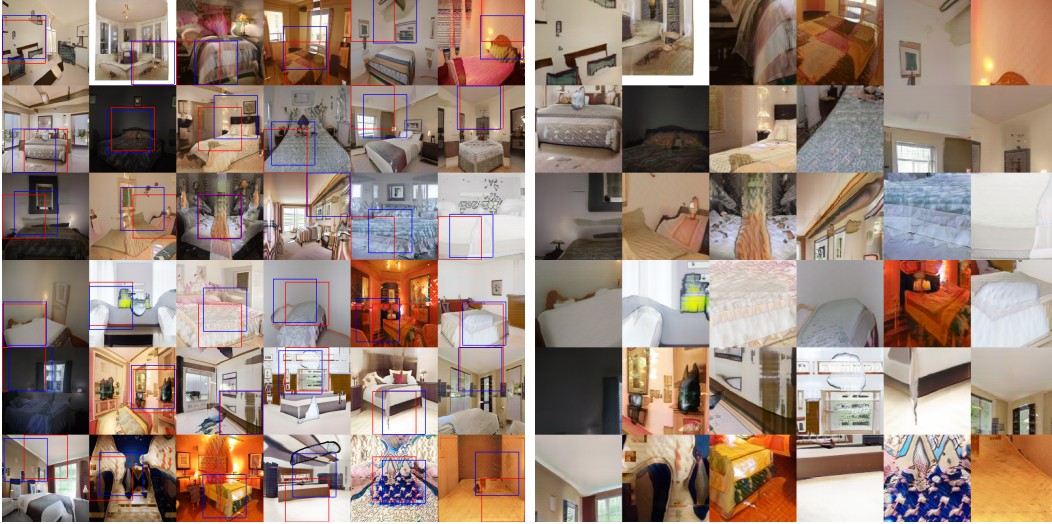

(c) (LSUN 256×256) Patch-inspired full-image generation.

(d) (LSUN 256×256) Patch-inspired macro patch generation.

Figure 20: Patch-inspired image generation can loosely retain some local structure or global characteristic of the original image. The red boxes are the sampled spatial positions that crop the full-images in (a) into the macro patches in (b). (c) & (d) show the patch-inspired generated macro patches and full-images based on $\tilde{z}$. The blue boxes visualize the predicted spatial position $\tilde{c}^m$. Since the information loss of the cropping process (from (a) to (b)) is critical, we do not expect (a) and (c) to be identical. Instead, (b) and (d) should be visually similar and (d) should be globally coherent. **Note that the diversity and difficulty of LSUN (bedroom category) is higher than CelebA. The $Q$ network can only capture the structure, shape, and orientation of the room and the bed, but it fails to capture detailed texture of objects**.

## APPENDIX H    TRAINING INDICATORS

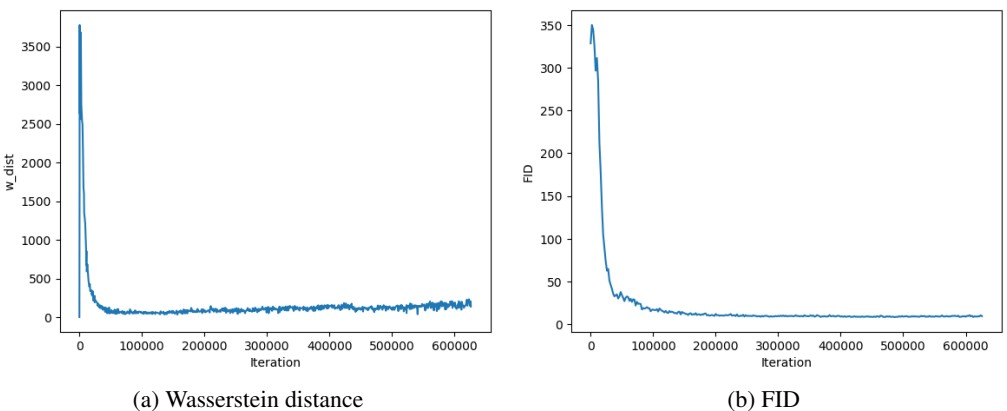

(a) Wasserstein distance

(b) FID

Figure 21: Both Wasserstein distance and FID through time show that the training of COCO-GAN is stable. Both two figures are logged while training on CelebA with $128 \times 128$ resolution.

## APPENDIX I    SPATIAL POSITIONS INTERPOLATION

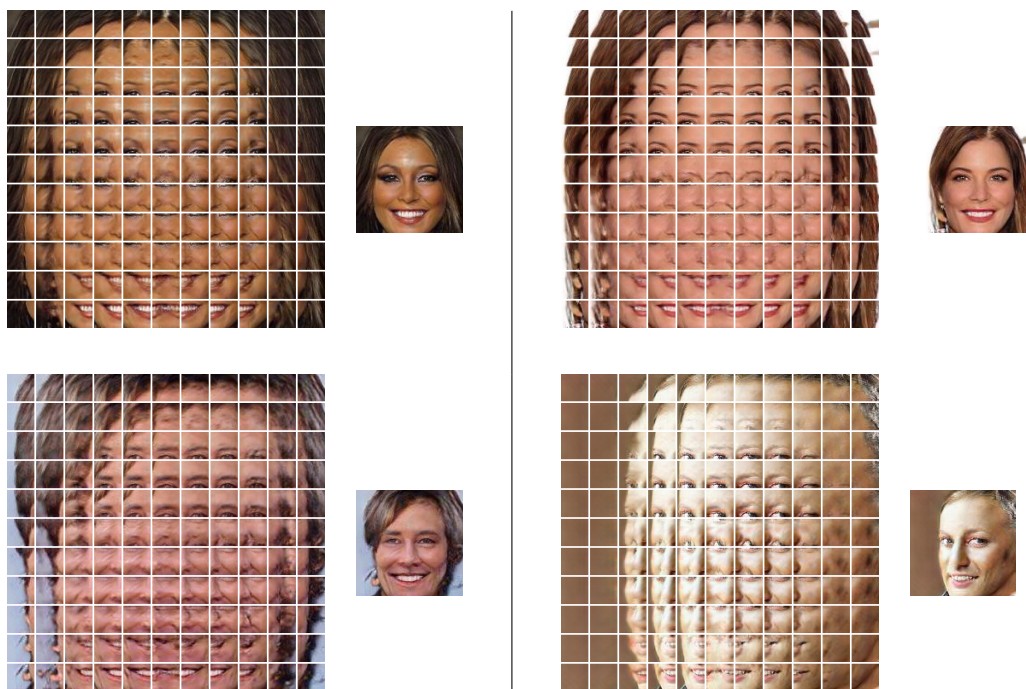

Figure 22: Spatial interpolation shows the spatial continuity of the micro patches. The spatial conditions are interpolated between range $[-1, 1]$ of the micro coordinate with a fixed latent vector.

