# OpenReview forum: "COCO-GAN: Conditional Coordinate Generative Adversarial Network"
_ICLR.cc/2019/Conference_

### Official Review · AnonReviewer1 · 2018-11-03
**Interesting idea but needs more work**

**Rating:** 4
**Confidence:** 5

**Review:**

This paper proposes to constrain the Generator of a WGAN-GP on patches locations to generate small images (“micro-patches”), with an additional smoothness condition so these can be combined into full images. This is done by concatenating micro-patches into macro patches, that are fed to the Discriminator.  The discriminator aims at classifying the macro-patches as fake or real, while additionally recovering the latent noise used for generation as well as the spatial prior.

There are many grammar and syntax issues (e.g. the very first sentence of the introduction is not correct (“Human perception has only partial access to the surrounding environment due to the limited acuity area of fovea, and therefore human learns to recognize or reconstruct the world by moving their eyesight.”). The paper goes to 10 pages but does so by adding redundant information (e.g. the intro is highly redundant) while some important details are missing

The paper does not cite, discuss or compare with the related work “Synthesizing Images of Humans in Unseen Poses”, by G. Lalakrishan et al. in CVPR 2018.

Page. 3, in the overview the authors mention annotated components: in what sense, and how are these annotated?
How are the patches generated? By random cropping?

Still in the overview, page 3, the first sentence states that D has an auxiliary head Q, but later it is stated that D has two auxiliary prediction heads.  Why is the content prediction head trained separately while the spatial one is trained jointly with the discriminator? Is this based on intuition or the result of experimentations?

What is the advantage in practice of using macro-patches for the Discriminator rather than full images obtained by concatenating the micro-patches? Has this comparison been done?

While this is done by concatenation for micro-patches, how is the smoothness between macro-patches imposed?

How would this method generalise to objects with less/no structure?

In section 3.4, the various statements are not accompanied by justification or citations. In particular, how do existing image pinpointing frameworks all assume the spatial position of remaining parts of the image is known?

How does figure 5 show that model can be misled to learn reasonable but incorrect spatial patterns?

Is there any intuition/justification as to why discrete uniform sampling would work so much better than continuous uniform sampling? Could these results be included?

How were the samples in Figure.2 chosen? Given that the appendix. C shows mostly the same image, the reader is led to believe these are carefully curated samples rather than random ones.

---

> ### Author Response · Authors · 2018-11-07
> **Response to AnonReviewer1 [3/3]**
>
>
> 11.
> “How were the samples in Figure.2 chosen? Given that the appendix. C shows mostly the same image, the reader is led to believe these are carefully curated samples rather than random ones.”
>
> > First, we only save 64 (8x8) randomly generated images for each epoch. Then we pick the epoch which has the lowest validation FID score. Since we believe showing fewer samples (which makes each generated sample larger) can make the generated samples clearer to see on the paper, so we just reuse and show the upper-left 25 samples.
> > We thank for the reviewer pointing out that our process may lead the reader to doubt that these samples are cherry-picked. We believe by:
> >     1. Release generated samples across epochs (each epoch with 64 images).
> >     2. Release the source code after acceptance.
> >     3. Replace the images in Figure. 2.
> >     4. The FID score also matches the image quality.
> > can support our samples are truly random-selected.
> >
> > The anonymous link to the per-epoch generated samples (CelebA 128x128 and LSUN 256x256):
> >   https://www.dropbox.com/sh/ucpthw2mnu3yw3g/AAC0AU5f7f1RfOvB3C5RM1YUa?dl=0

---

> ### Author Response · Authors · 2018-11-07
> **Response to AnonReviewer1 [2/3]**
>
>
> 6.
> "While this is done by concatenation for micro-patches, how is the smoothness between macro-patches imposed?"
>
> > The smoothness between patches is taken care by the adversarial loss. The discriminator oversees whether the concatenated patches have discontinuities between the concatenated edges. In the meanwhile, the provided real samples have no such discontinuity. The adversarial loss guides the generator to match the real samples distribution, thus increasingly imposes the smoothness between the patches.
> > The explanation can be supported by a series of generated samples through time. We attach a series of generated full images through time in the anonymous link below. Especially for the CelebA 128x128 setting (note that LSUN has much more iterations each epoch), we can observe the seam between patches fades away rapidly through time.
> > The link to the per-epoch generated samples (CelebA 128x128 and LSUN 256x256):
> >   https://www.dropbox.com/sh/ucpthw2mnu3yw3g/AAC0AU5f7f1RfOvB3C5RM1YUa?dl=0
>
> 7.
> “How would this method generalise to objects with less/no structure?”
>
> >   If we understand correctly, by “objects with structure” you mean the CelebA and LSUN are having strong structure priors (e.g. eyes positions of CelebA and bed position of LSUN).
> > We believe the panoramas generated in Figure 6 & 7 are relatively unstructured data. Any item within the panorama does not have deterministic patterns of its position. COCO-GAN works reasonably nice with the panorama dataset in a cylindrical coordinate system and should work with most of the common image datasets without significant structure priors assumed.
>
> 8.
> "In section 3.4, the various statements are not accompanied by justification or citations. In particular, how do existing image pinpointing frameworks all assume the spatial position of remaining parts of the image is known?"
>
> > We assume there is a typo and the reviewer is referring to image “inpainting” frameworks.
> > Thanks for pointing this problem out, we will modify our statements and properly cite papers.
> > In many cases of real-world applications, some categories of damaged images do not preserve their spatial position in its original full-image (e.g., corrupted or cropped-out photos). Considering these cases, common image inpainting frameworks \cite{A,B,C} only consider the remaining parts of the image are already in their optimal positions. In comparison, the discriminator of COCO-GAN is trained by $L_{S}$ to predict the expected placement of the given macro patch.
>
> > [A] Image Inpainting for Irregular Holes Using Partial Convolutions
> > [B] Semantic Image Inpainting with Deep Generative Models
> > [C] High-Resolution Image Inpainting using Multi-Scale Neural Patch Synthesis
>
> 9.
> “How does figure 5 show that model can be misled to learn reasonable but incorrect spatial patterns?”
>
> > Thanks for pointing out that we should reference this statement to “Section 3.2, paragraph two of Spatial Positions Interpolation”.
> > In Figure 5, let (x, y) be the zero-based position of the patch counting from top-to-bottom (x-axis) and left-to-right (y-axis). The patch at (4, 5) in both samples is expected to be a smooth area for a glabella between eyes. But in the generated full image, the generator is misled by the discrete spatial position sampling strategy. The generator learns to transform the shape of the eye to switch from one eye to another. This is a reasonable behavior due to the sparse sampling but is an incorrect pattern.
>
> 10.
> “Is there any intuition/justification as to why discrete uniform sampling would work so much better than continuous uniform sampling? Could these results be included?”
>
> > We can include the experimental results in the next paper revision. To ensure the results are not affected by any versioning problem, we will rerun the experiment in CelebA 64x64 setting, which will take couples of days to complete.
> > Aside from the experimental observation, an empirical explanation is that only discrete spatial positions are used during inference. Directly optimizing on these discrete spatial positions can surely result in better inference time generation quality. However, it would take generalization as the trade-off.

---

> ### Author Response · Authors · 2018-11-07
> **Response to AnonReviewer1 [1/3]**
>
> We sincerely appreciate the reviewer's effort on providing many useful comments in our paper. All reviewers agree our work is novel, which is our target to introduce the new “conditional coordinate” idea to the community. About the writing parts, we will update and reorganize the paper as soon as possible. And here are some responses to the reviewer's concerns and questions:
>
> 1.
> "There are many grammar and syntax issues (e.g. the very first sentence of the introduction is not correct (“Human perception has only partial access to the surrounding environment due to the limited acuity area of fovea, and therefore human learns to recognize or reconstruct the world by moving their eyesight.”). The paper goes to 10 pages but does so by adding redundant information (e.g. the intro is highly redundant) while some important details are missing"
>
> > Thanks for pointing some of the writing problems, we will upload a revised version in the following few days.
>
> 2.
> "The paper does not cite, discuss or compare with the related work “Synthesizing Images of Humans in Unseen Poses”, by G. Lalakrishan et al. in CVPR 2018."
>
> > We will accordingly mention this paper as an interesting related domain. However, COCO-GAN is not related to and comparable with the paper. We do not introduce any specific type of human or object prior to the model during training. Our solution is generalized to any type of image and it simply slices the input image into patches. Such prior can generalize to most of existing image types. This is related to 7. below.
>
> 3.
> "Page. 3, in the overview the authors mention annotated components: in what sense, and how are these annotated? How are the patches generated? By random cropping? "
>
> > We are not entirely sure if we catch the reviewer’s question correctly. We assume the reviewer is asking about “why we do not specify the patches sampling strategy in Page. 3.”
> > We do not specify how the patches are selected in this section since it is related to the characteristics of the dataset (e.g., panorama can be trained in a cylindrical coordinate system, and one can sample patches which cross the left and right edges) and may have many different implementations as long as the patches are sampled nearby (e.g., any NxM patches are acceptable, as long as the computational budget is sufficient).
> > We define the micro/macro coordinates and patches used in the experiment in ``the first paragraph of experiment section'' and ``the second paragraph of section 3.3''. The former one is a straightforward version validating the generation quality, while the latter one applies COCO-GAN to the cylindrical coordinate system for panoramas.
>
> 4.
> "Still in the overview, page 3, the first sentence states that D has an auxiliary head Q, but later it is stated that D has two auxiliary prediction heads.  Why is the content prediction head trained separately while the spatial one is trained jointly with the discriminator? Is this based on intuition or the result of experimentations?"
>
> > This is because the content prediction head is designed based on info-GAN, where a  Q network is trained separately. In the meanwhile, the spatial prediction head is similar to ACGAN, which is trained jointly in its original implementation. We decide to optimize these two losses with their original strategies.
>
> 5.
> "What is the advantage in practice of using macro-patches for the Discriminator rather than full images obtained by concatenating the micro-patches? Has this comparison been done?"
>
> > The discriminator with full image observation will surely lead to better full image generation quality. However, two of our three applications and benefits listed in the introduction require the discriminator to be trained with macro-patches.
> >  For “Patch-Inspired Image Generation”, the discriminator needs to learn a mapping from each macro patch to its original latent vector and spatial position.
> >  For “Computation-Friendly Generation”, although we only describe the benefits for the generator in the inference stage, however, the discriminator may also need patch-based training if the training process also reaches memory budget limit (e.g., for extremely large image generation or the model has relatively more parameters).
> > We will follow up with an experiment which makes the discriminator training with the full image in CelebA 64x64 setting. But as a side effect, we will remove $L_{S}$ since the discriminator lacks a macro coordinate system.

---

> ### Author Response · Authors · 2018-11-25
> **Rebuttal update**
>
> Thanks for all the reviewers’ effort in the paper review, we received lots of valuable suggestions. We accordingly revised our paper and create a meta-summary of the rebuttal. Please kindly check it out and leave some comments. We are more than willing to discuss more and further polish our paper in the remaining rebuttal period!
>
> Link to the meta-summary by the author:
> https://openreview.net/forum?id=r14Aas09Y7&noteId=ryxR2VUORX

---

### Official Review · AnonReviewer2 · 2018-11-03
**interesting idea that surprisingly works**

**Rating:** 6
**Confidence:** 4

**Review:**

+ Interesting and novel idea
+ It works
- Insufficient ablation and comparison
- Unclear what the advantages of the presented framework are

The presented idea is clearly new and a deviation from standard GAN architectures. I was surprised to see that this actually produces visually coherent results. I was certain that it would create ugly seams at the boundary. For this reason I like the submission overall.

However, the submission has two major short-comings. First, throughout the exposition it is never really clear why COCO-GAN is a good idea beyond the fact that it somehow works. I was missing a concrete use case where COCO-GAN performs much better.

Second, I was missing any sort of ablation experiments. The authors only evaluate the complete system, and never show which components are actually needed. Specifically, I'd have liked to see experiments:
 * with/without a context model Q
 * with a standard discriminator (single output or convolutional), but a micro-coordinate generator
 * with a macro-block discriminator, but a standard generator
 * without coordinate conditioning, but different Generator parameters for each coordinate

These experiments would help better understand the strength of COCO-GAN and how it fits in with other GAN models.

Minor:
The name of the method is not ideal. First, it collides with the COCO dataset. Second, it does not become clear why the proposed GAN uses a "Conditional Coordinate until late in the exposition. Third, the main idea could easily stand without the coordinate conditioning (see above).

---

> ### Author Response · Authors · 2018-11-07
> **Response to AnonReviewer2**
>
> Sincerely thanks for the valuable suggestions from the reviewer. All reviewers agree our work is novel, which is our target to introduce the new “conditional coordinate” idea to the community. Here are some responses to the reviewer’s question:
>
> 1.
> “The presented idea is clearly new and a deviation from standard GAN architectures. I was surprised to see that this actually produces visually coherent results. I was certain that it would create ugly seams at the boundary. For this reason I like the submission overall.”
>
> > Thanks for being interested in one of our most important observations. We believe this characteristic provides many merits to different tasks, which is our main thread across all analysis, discussion and experiments.
>
> 2.
> “However, the submission has two major short-comings. First, throughout the exposition it is never really clear why COCO-GAN is a good idea beyond the fact that it somehow works. I was missing a concrete use case where COCO-GAN performs much better.”
>
> >  We will update the introduction section in the following days to ensure the reader can have a fast and clear view of the main contributions and use cases of COCO-GAN.
> > In general, we first observe COCO-GAN has fewer seams than we expected. This property enables both G and D to learn with partial views (i.e., micro patches and macro patches, respectively). We believe this non-before-seen property has three interesting merits:
> >     1. Patch-Inspired Image Generation
> >     2. Partial-Scene Generation
> >     3. Computation-Friendly Generation.
> > We further discuss and perform experiments to support these applications and benefits.
>
> 3.
> “Second, I was missing any sort of ablation experiments. The authors only evaluate the complete system, and never show which components are actually needed. Specifically, I'd have liked to see experiments:
>  * with/without a context model Q
>  * with a standard discriminator (single output or convolutional), but a micro-coordinate generator
>  * with a macro-block discriminator, but a standard generator
>  * without coordinate conditioning, but different Generator parameters for each coordinate”
> “These experiments would help better understand the strength of COCO-GAN and how it fits in with other GAN models.”
>
> > Thanks for  pointing this out. We agree although each component of COCO-GAN is necessary for specific applications in our work, some users may not necessarily need all applications or benefits at the same time.
> > We will perform the former three ablation studies in the following days in CelebA 64x64 setting, which is relatively fast, and also other datasets afterward.
> > However, the last one is slightly out-of-topic. This will result in a dramatic increase in the total number of parameters. In our basic setting, we split the full image into 4x4 micro patches, and 12x4 micro patches for panorama dataset. The suggested setting of the last ablation study might not a feasible solution to real-world applications. Furthermore, it is hard to perform a fair comparison between models with different numbers of total parameters and FLOPs.
>
> 4.
> “The name of the method is not ideal. First, it collides with the COCO dataset. Second, it does not become clear why the proposed GAN uses a "Conditional Coordinate until late in the exposition. Third, the main idea could easily stand without the coordinate conditioning (see above).”
>
> > We are aware of this problem. Conditional coordinate is our core idea and component. We believe it is important and should appear in the name of model. We will update the introduction to make the idea clearer.
> > Lastly, we believe conditional coordinate is essential for our framework. The generator learns to generate micro patches based on their coordinate and take cares of edge smoothness with respect to their potential siblings, which are also defined by the coordinate system. Furthermore, generating panorama in the cylindrical coordinate system is also a nature and straightforward choice, and investigating other coordinate systems for different image types is also an interesting and unexplored research direction.

---

> > ### Comment · AnonReviewer2 · 2018-11-08
> > **RE: Response to AnonReviewer2**
> >
> > Thank you for the quick reply.
> > 2. Can you show some concrete experiments supporting these claims?
> >
> > 3. I still think ex 4 is very important. I understand that you'll have many more parameters, but the computation should be the same (or slightly smaller, since you do not have the conditional coordinate). However, only this experiment highlights the need for coordinate conditioning (which according to reply 4 is the essential component of this paper).

---

> > > ### Author Response · Authors · 2018-11-10
> > > **Response to AnonReviewer2**
> > >
> > > Thanks for keeping in touch with us! Here are our responses:
> > >
> > > 2.
> > > “Can you show some concrete experiments supporting these claims?”
> > >
> > >
> > > ** Patch-Inspired Image Generation **
> > > Section 3.4, Figure 8 and Appendix E are related to this application.
> > >
> > > ** Partial-Scene Generation **
> > > This is mentioned in the last paragraph of Section 3.3. Generating partial views of CelebA or LSUN may not have real-world applications, but generating part of panoramas in virtual reality (VR) does have prospective benefits in reducing computations. We show that COCO-GAN can generate panorama in a cylindrical coordinate system. Although the full panorama resolution in our experiment is not significantly high, we believe our experiment is a valid proof of concept.
> > >
> > > ** Computation-Friendly Generation **
> > > Section 3.5 discusses this issue and our experiments support that the idea of generating images via concatenating multiple generated patches is possible.
> > >
> > > 3.
> > > “I still think ex 4 is very important. I understand that you'll have many more parameters, but the computation should be the same (or slightly smaller, since you do not have the conditional coordinate). However, only this experiment highlights the need for coordinate conditioning (which according to reply 4 is the essential component of this paper).”
> > >
> > > We have kick-started this experiment under CelebA 64x64 setting.
> > > One of the critical problems of this solution is that, take the panorama generation setting for example, this solution will need to fit 48 different generators into the GPU memory during training phase. The scale of this problem grows rapidly as the image size increases. Although there may exist some engineering workarounds to make this possible, but when a weight-sharing strategy like COCO-GAN exists, we believe COCO-GAN will be more attractive.

---

> > > ### Author Response · Authors · 2018-11-25
> > > **Rebuttal update**
> > >
> > > Thanks for all the reviewers’ effort in the paper review, we received lots of valuable suggestions. We accordingly revised our paper and create a meta-summary of the rebuttal. Please kindly check it out and leave some comments. We are more than willing to discuss more and further polish our paper in the remaining rebuttal period!
> > >
> > > Link to the meta-summary by the author:
> > > https://openreview.net/forum?id=r14Aas09Y7&noteId=ryxR2VUORX

---

> > > > ### Comment · AnonReviewer2 · 2018-11-26
> > > > **LGTM, remove Q network from technical section**
> > > >
> > > > I just looked over the revision and I'm happy with the changes and additional results.
> > > >
> > > > I'd recommend the authors to remove the Q-network from the main technical section for the final revision. It does hurt performance (contributes negatively). If the authors still want to talk about it, the appendix might be a better place for it.

---

> > > > > ### Author Response · Authors · 2018-11-27
> > > > > **Thanks for support**
> > > > >
> > > > > We agree with the suggestion from the reviewer. Since the Q-network is highly correlated to the “Patch-Inspired Image Generation” application, we would move it to the corresponding experiment section with a brief description and leave the details in the appendix for the final revision for better readability.

---

### Official Review · AnonReviewer3 · 2018-11-07
**Interesting Ideas, but not validated**

**Rating:** 6
**Confidence:** 4

**Review:**

The paper describes a GAN architecture and training methodology where a generator is trained to generate "micro-" patches, being passed as input a latent vector and patch co-ordinates. Micro-patches generated for different adjacent locations with the same latent vector are combined to generate a "macro" patch. This "macro" output is trained against a discriminator that tries to label this output as real and fake, as well as predict the location of the macro patch and the value of the latent vector. The generator is trained to fool the discriminator's label, and minimize the error in the prediction of location and latent vector information.

- The paper proposes a combination of different interesting strategies. However, a major drawback of the method is that it's not clear which of these are critical to the quality of the generated output.

- Firstly, it isn't clear to me why the further breakdown of the macro patch into micro patches is useful. There appears to be no separate loss on these intermediate outputs. Surely, a DC-GAN like architecture with sufficient capacity would be as well able to generate "macro" patches. The paper needs to justify this split into micro patches with a comparison to a direct architecture that generates the macro patches (everything else being the same). Note that applications like "interpolation" of micro patches could be achieved simply by interpolating crops of the macro patch.

- As a means of simply producing high-resolution images, it appears that "PGGAN" performs better than the proposed method. Therefore, the paper doesn't clearly explain the setting when the division into patches produces a better result. It is worth noting that the idea of applying "local" critics (i.e., discriminators acting on sub-regions) isn't new (e.g., Generative Image Inpainting with Contextual Attention in CVPR 2018). What's new is the proposed method's way of achieving consistency between different regions by providing the 'co-ordinate' of the patch as input (and seeking consistency in the latent vector through a loss)---rather than applying a discriminator at a coarser level on the downsampled image. But given the poorer performance compared to PGGAN, it isn't clear that there is a an advantage to this approach.

Overall, the paper brings up some interesting ideas, but it doesn't motivate all its design choices, and doesn't make a clear argument about the settings in which the proposed method would provide an actual advantage.

===Post-rebuttal

I'm upgrading my score from 5 to 6, because some of the ablation experiments do make the paper stronger. Having said that, I still think this is a borderline paper. "Co-ordinate conditioning" is an interesting approach, but I think the paper still lacks convincing experiments for its main motivating use case: generating outputs at a resolution that won't fit in memory within a single forward pass. (This motivation wasn't clear in the initial version, but is clearer now).

The authors' displayed some high-resolution results during the rebuttal phase, but note that they haven't tuned the hyper-parameter for these (and so the results might not be the best they can be). Moreover, they scale up the sizes of their micro and macro patches so that they're still the same factor below the full image. I think a version of this paper whose main experimental focus is on high-resolution data generation, and especially, from much smaller micro-macro patches, would make a more convincing case.

So while the paper is about at the borderline for acceptance, I do think it could be much stronger with a focus on high-resolution image experiments (which is after all, forms its motivation).

---

> ### Author Response · Authors · 2018-11-08
> **Response to AnonReviewer3**
>
> We sincerely appreciate the reviewer raises many important questions we are more than willing to discuss. All reviewers agree our work is novel, which is our target to introduce the new “conditional coordinate” idea to the community. Note that since some questions are correlated, our response is not in the reviewer’s original question order.
>
> 1.
> “As a means of simply producing high-resolution images, it appears that "PGGAN" performs better than the proposed method. Therefore, the paper doesn't clearly explain the setting when the division into patches produces a better result. It is worth noting that the idea of applying "local" critics (i.e., discriminators acting on sub-regions) isn't new (e.g., Generative Image Inpainting with Contextual Attention in CVPR 2018). What's new is the proposed method's way of achieving consistency between different regions by providing the 'co-ordinate' of the patch as input (and seeking consistency in the latent vector through a loss)---rather than applying a discriminator at a coarser level on the downsampled image. But given the poorer performance compared to PGGAN, it isn't clear that there is a an advantage to this approach.”
>
> > The main target of COCO-GAN is exploring other applications instead of increasing generation quality. The low FID score is a by-product.
> > In many real-world applications (e.g., VR, medical images), the data are normally too large to even fit into memory. Modern GAN architectures require the generator to generate the full image at once. This requirement makes generating these super-large images hard to achieve. As the result, we propose COCO-GAN, which can break full-image generation into patches generation. Furthermore, the discriminator also takes macro patches as input, since taking the full image as the input of the discriminator is infeasible in super-large image generation problem.
> > COCO-GAN is orthogonal to PGGAN, one can still add the “progressive growing” strategy to micro/macro patches generation/discrimination. However, this will introduce more hyperparameters, thus making it more challenging to balance everything.
>
> 2.
> “Firstly, it isn't clear to me why the further breakdown of the macro patch into micro patches is useful. There appears to be no separate loss on these intermediate outputs. Surely, a DC-GAN like architecture with sufficient capacity would be as well able to generate "macro" patches. The paper needs to justify this split into micro patches with a comparison to a direct architecture that generates the macro patches (everything else being the same). Note that applications like "interpolation" of micro patches could be achieved simply by interpolating crops of the macro patch.”
>
>
> > The output of generator *must* be smaller than the input of the discriminator for COCO-GAN. This is for smoothening the seam between patches after concatenating multiple patches. The discriminator oversees whether the concatenated patches have discontinuities between the concatenated edges.
> > In the CelebA 128x128 setting, our micro patches are of size 32x32, macro patches are of size 64x64. Even if the generator generates macro patches, it still needs to take care of seams while producing the full image. If one decides to make the discriminator taking the full image as input while the output of generator is a 64x64 patch, which can be done via concatenating four patches produced by the generator, then it becomes another special case of COCO-GAN.
> > We provide an anonymous link below. The seam between patches is smoothed out  through time. This suggests the adversarial loss takes cares of the seam between patches.
> > As described in response to 3., we will update the methodology section to justify this design.
> >
> > The anonymous link to the per-epoch generated samples (CelebA 128x128 and LSUN 256x256):
> >   https://www.dropbox.com/sh/ucpthw2mnu3yw3g/AAC0AU5f7f1RfOvB3C5RM1YUa?dl=0
>
>
> 3.
> “Overall, the paper brings up some interesting ideas, but it doesn't motivate all its design choices, and doesn't make a clear argument about the settings in which the proposed method would provide an actual advantage.”
>
> > We will update the methodology section to discuss the design motivations of each component and the introduction section to make arguments of actual advantages more clear.

---

> > ### Comment · AnonReviewer3 · 2018-11-09
> > **Follow-up**
> >
> > Thanks for the reply.
> >
> > So, it seems like the main motivation or use-case is when it wouldn't be possible to generate entire images (or volumes) at a time because the a single generator for the entire image doesn't fit in memory.
> >
> > But it seems the straight-forward solution in that case would be to simply take crops of intermediate feature maps in the generator. For example, to generate a specific crop of a full-sized image with a typical progressive generator (that increases resolution with upsampled convolution layers), you could crop every intermediate layer's activations so as to only retain the portion that is part of the receptive field of the final output.
> >
> > This is the normal approach to processing large scale inputs/outputs. Passing co-ordinates as input and then trying to achieve consistency post-facto seems to be un-necessary if memory constraints are the only issue.
> >
> > Note that you could apply this strategy during training as well if memory was an issue (choosing random final image-space crops, and cropping accordingly before passing to the discriminator).

---

> > > ### Author Response · Authors · 2018-11-10
> > > **Thanks for the follow-up!**
> > >
> > > Sincere thanks for the follow-up and suggestions, which are helpful for further highlighting the advantages of our method:
> > >
> > > At first glance, a potential problem of the method (referred as [M] afterward) proposed by the reviewer is in the normalization layer. Most of the common normalization methods do not work with partial feature maps. A possible solution to this problem is using conditional batch normalization similar to which used in our paper. This makes [M] very similar to COCO-GAN as it introduces spatial conditions during generation, except [M] has much more parameters.
> > >
> > > Another implementation difficulty is that [M] needs to consider padding of convolutional layers. The cropped feature map in consecutive layers is not trivially 1x or 2x sized. Let:
> > >    S: the size of the next feature map
> > >    P: the padding required
> > >    F: the fraction of down-scaling,
> > > the previous feature map requested is F*L+P, and P needs to consider paddings if the patch is occasionally near the edge of the feature map. COCO-GAN is relatively easy to implement and flexible to different architectures.
> > >
> > > Here’s a brief comparison of pros and cons between COCO-GAN and [M] from our point of view:
> > >
> > > COCO-GAN:
> > >  + Significantly fewer parameters
> > >  + Considers coordinate system, which shows appealing results in our panorama setting, and may further promote to other image types (such as 360 images with a hyperbolic coordinate system)
> > >  + The generation quality is surprisingly good, no obvious seams without post-processing.
> > >  + Flexible to different generator architectures.
> > >  - Need to define coordinate systems and patch size, which can be treated as a set of hyper-parameters.
> > >  - Slight but reasonable FID score drop relative to optimal model (generate and discriminate on full-image)
> > >
> > > [M]:
> > >  + Straightforward
> > >  - Friendly to dynamic graph frameworks, but relatively hard to implement for static graph frameworks (e.g. Tensorflow)
> > >  - To our knowledge, no related GAN publications or analysis about the potential problems.
> > >  - The normalization layer may need to adopt conditional batch norm, which makes it similar to COCO-GAN
> > >  - The generator and the discriminator are significantly imbalanced. This may cause the model hard to train in practice.
> > >  - The discriminator may need spatial positions as condition to form a cGAN or ACGAN loss, which is again similar to COCO-GAN.
> > >
> > > We hope our empirical analysis above is justifiable for the reviewer to agree that COCO-GAN has its value and advantages. Although [M] could also be a possible solution, it would require further study. Besides, given the tight time constraint of rebuttal period, it is hard to implement, debug, train and fine-tune [M] to sufficiently good quality to compare with COCO-GAN.

---

> > > > ### Comment · AnonReviewer3 · 2018-11-10
> > > > **Not clear why normalization would be an issue**
> > > >
> > > > It's not clear why you'd need conditional batch-norm or why it would be an issue to implement in a static graph framework like Tensorflow. You could simply add a tf.slice (with fixed size crop for each layer, but random co-ordinates) after the output of every conv2d_transpose layer, and then do regular batch-normalization on that sliced output.
> > > >
> > > > Note that in the proposed method, batch-normalization is also being done on a smaller sized feature maps (corresponding to each crop)---it would be the same in this case, except that those smaller sized feature maps would come from cropping a larger map.
> > > >
> > > > Note that working with crops is a fairly common solution in a lot of other applications which produce full-sized maps like segmentation, super-resolution, depth estimation, etc.  (although there the original input itself is randomly cropped) even though its not been used for GANs before.
> > > >
> > > > At some level, cropping (that authors refer to as method [M]) is essentially the same as what R2 also asked for in their review---i.e., separate generators for each co-ordinate. [M] is basically a convolutional version of that (i.e., each crop of the full generator is essentially a separate generator for that  crop's co-ordinates).

---

> > > > > ### Author Response · Authors · 2018-11-12
> > > > > **Thanks for the rapid response [2/2]**
> > > > >
> > > > >
> > > > > 2.
> > > > > “Note that working with crops is a fairly common solution in a lot of other applications which produce full-sized maps like segmentation, super-resolution, depth estimation, etc.  (although there the original input itself is randomly cropped) even though its not been used for GANs before.”
> > > > >
> > > > >
> > > > > Yes, to our knowledge in the GANs domain, we are the first work observing the phenomenon that the generated partial patches can be concatenated without post-processing and result in surprisingly high-quality images. Furthermore, our setting is very different from existing patch-based solutions for segmentation, super-resolution, depth estimation. First of all, those methods mostly have very strong semantic and structure self-similarity between inputs and outputs, which makes the result patches easy to align. Second, segmentation is relatively easy to retain continuity between patches, as they are not output in raw pixel space. Third, the patches generated by COCO-GAN are entirely non-overlapped, while normal patch-based approaches consider calculating the mean of multiple overlapped patches. Lastly, even the same setting in different domains have different problems to deal with. We propose, implement, analyze and verify the idea of  COCO-GAN, and we show that it is a valid solution after overcoming many unexplored problems. In the belief that these are interesting observations and COCO-GAN may open many new possible applications, we are excited about sharing this work with the community.
> > > > >
> > > > >
> > > > >
> > > > > 3.
> > > > > “At some level, cropping (that authors refer to as method [M]) is essentially the same as what R2 also asked for in their review---i.e., separate generators for each co-ordinate. [M] is basically a convolutional version of that (i.e., each crop of the full generator is essentially a separate generator for that  crop's co-ordinates).”
> > > > >
> > > > > We are running the experiment of training multiple generators based on this rebuttal request, which we expect may cause the optimization process to become slower due to a significant increase in the number of parameters. Just to make things more precise: COCO-GAN shares weights across different spatial positions and has fewer parameters. In contrast, training multiple generators introduces significantly more parameters, and [M] also needs to increase the number of parameters as more convolutional layers are needed.
> > > > >
> > > > > Lastly, the multiple generators setting and [M] are both a special case of COCO-GAN except introducing more parameters. The former one explicitly introduces conditional coordinates via generator selection. The latter one implicitly introduces spatial conditional coordinate via selecting specific slices of feature maps.
> > > > >
> > > > > We thank the reviewers for suggesting possible solutions to our original objective. The three methods (including COCO-GAN) have different pros and cons. Further empirical comparisons between these approaches may be an interesting future work direction, but probably not in the scope of this paper as we aim to introduce conditional coordinate (an explicitly or implicitly used feature for all three methods), exemplify the generation quality (while [M] needs some further work to justify) and show some of (but not limited to) the possible new applications.

---

> > > > > > ### Comment · AnonReviewer3 · 2018-11-12
> > > > > > **Response**
> > > > > >
> > > > > > - So for the static graph generation, I think there may have been a mis-understanding. You don't need to generate a graph for all the crops (for training or generation). You would generate the graph for one crop (with the location of the crop being a placeholder at test time, or randomly generated for training). For generation, you would call the same graph multiple times with the same latent vector and different crop locations, and then just concatenate the results on the CPU---just like you're calling this paper's generator with different co-ordinates and concatenating them together. Training would be with one random crop at a time.
> > > > > >
> > > > > > - About requiring conditional batch normalization (although, you could have different crops for different members in a batch---just like you have different co-ordinates in the current setting) and differences in parameters, yes, these are empirical questions---but that's why experiments would help to resolve this. It is possible that a cropping strategy may not work for GANs as it does for other tasks---or at least not as well as the proposed method, but again, experimental validation for this would be good.
> > > > > >
> > > > > > - I think more generally, the main issue with the experimental validation is that the method is motivated by the need to generate very high resolution volumes, but the experiments are run on standard small datasets (the panoramas are examples of stitching things together to being moderately large outputs---but these weren't actually trained on high-resolution panoramas, so it is difficult to figure out whether they are plausible).
> > > > > >
> > > > > > I think a lot of the requests from reviewers for ablation and baselines would go away if the method were convincingly able to learn generation for truly high-dimensional images or signals---the kind where traditional GANs would run out of memory and clearly could not be used. That would be definitively demonstrate that method is able to "achieve" something that prior work wasn't able to.
> > > > > >
> > > > > > Without that kind of validation, I think the paper needs to run more of these baseline experiments to make its case.

---

> > > > > > > ### Author Response · Authors · 2018-11-14
> > > > > > > **Author's response [2/2]**
> > > > > > >
> > > > > > > “- I think more generally, the main issue with the experimental validation is that the method is motivated by the need to generate very high resolution volumes, but the experiments are run on standard small datasets (the panoramas are examples of stitching things together to being moderately large outputs---but these weren't actually trained on high-resolution panoramas, so it is difficult to figure out whether they are plausible).”
> > > > > > >
> > > > > > > We validate that our method can learn to compose larger images with patches. Our method is compatible with most of the mainstream GAN methods. One can combine COCO-GAN with PGGAN without conflicts (except may require massive computing power for hyperparameters tuning) since COCO-GAN only affects the input/output of the model and introduces CBN to replace batch norm.
> > > > > > >
> > > > > > > We think, at this point, it would be helpful to validate that COCO-GAN can still generate high-quality images even in the following cases:
> > > > > > >     1. Directly concatenating the generated patches without observing extreme seams.
> > > > > > >     2. Both the generator and the discriminator have only partial observations.
> > > > > > >
> > > > > > > About validating generation quality directly on the high-resolution dataset, we decide to provide an anonymous link to a set of 1024x1024 samples generated by COCO-GAN. We run this experiment with our default hyperparameter configuration (this config was obtained from CelebA 64x64 experiment) on CelebA-HQ with 256x256 resolution patches.
> > > > > > >
> > > > > > > We are hesitant to provide these images as we are afraid if the community may have an unfair impression about  COCO-GAN’s generation quality. So far this experiment requires more hyperparameter tuning and the training may still not converge yet. We observe significant balancing problem in the G/D loss curve. Note that hyperparameter tuning for CelebA-HQ is very expensive and essential for GANs, but we are not affordable for that. PGGAN takes 2 weeks to train on CelebA-HQ, meanwhile, COCO-GAN should converge slower than PGGAN, since it does not adopt the “progressive growing” strategy.
> > > > > > >
> > > > > > > Anonymous link to CelebA-HQ 1024x1024 generation samples:
> > > > > > > https://www.dropbox.com/sh/5ly8xk22cqhxt76/AAAee1E2D8rIPAwFZtympnnta?dl=0
> > > > > > >
> > > > > > > Note that the spatial interpolation property does not exist in [M]. We believe the most important contribution of this work is whether our observation and some non-before-seen properties of COCO-GAN are interesting to the community.
> > > > > > >
> > > > > > > “I think a lot of the requests from reviewers for ablation and baselines would go away if the method were convincingly able to learn generation for truly high-dimensional images or signals---the kind where traditional GANs would run out of memory and clearly could not be used. That would be definitively demonstrate that method is able to "achieve" something that prior work wasn't able to.
> > > > > > > Without that kind of validation, I think the paper needs to run more of these baseline experiments to make its case.”
> > > > > > >
> > > > > > > We believe COCO-GAN is technically a favorable solution toward high-resolution image generation. We provide 1024x1024 resolution generation, which is already on the state-of-the-art level in terms of resolution, though it still needs some hyperparameter tuning. We believe this is a direct evidence that COCO-GAN can apply to high-resolution image generation. If given sufficient computing power, more extensive hyperparameter tuning, and maybe combined with some mainstream GANs training frameworks (such as PGGAN), we are confident to show that COCO-GAN is an important building block for high-resolution image generation.
> > > > > > >
> > > > > > > Note that we are running the experiments for the five rebuttal requests from R1 and R2, it still needs to take some time to wait until the models converge.

---

> > > > > > > > ### Comment · AnonReviewer3 · 2018-11-19
> > > > > > > > **Response**
> > > > > > > >
> > > > > > > > - Agree to limit discussion of [M] (I think there's still a misunderstanding, I wasn't suggesting generating different graphs on the fly, but a single graph for each crop, and providing crop locations as tensorflow placeholders). I believe that the comparisons to the per-coordinate generators requested by the other reviewer will be a good enough equivalent baseline.
> > > > > > > >
> > > > > > > > - About the high-resolution results: I think these results are important, since this is the main motivating use case of COCO-GAN. We saw a drop in scores compared to PGGAN at the lower resolution. The question is that does that drop become larger when one goes to higher resolution---i.e., is a single generator with co-ordinate input still able to generate the whole image. As you say, fewer parameters is better, but only if they still give comparable performance.
> > > > > > > >
> > > > > > > > Question: for 1024x1024 full images, are you using 256x256 as the resolution of the micro patches ? What happens if you use the same hyper-parameters for the 128x128 dataset (i.e., 32x32 micro-patches) and use it to generate the 1024x1024 images ?
> > > > > > > >
> > > > > > > > This is again important in the context of the motivation of memory usage. Given a size of micro-patches that can be fit in memory, what factor larger images can one generate using COCO-GAN ? Is it limited to only 4x ?

---

> > > > > > > > > ### Author Response · Authors · 2018-11-19
> > > > > > > > > **Thanks for the questions!**
> > > > > > > > >
> > > > > > > > > “- Agree to limit discussion of [M] (I think there's still a misunderstanding, I wasn't suggesting generating different graphs on the fly, but a single graph for each crop, and providing crop locations as tensorflow placeholders). I believe that the comparisons to the per-coordinate generators requested by the other reviewer will be a good enough equivalent baseline.”
> > > > > > > > >
> > > > > > > > > We have finished a straightforward implementation of [M]. Although it already shows significant differences in quality between [M] and COCO-GAN, considering the model training is still not fully-converged, we will still wait for a couple of days to monitor it.
> > > > > > > > >
> > > > > > > > > “- About the high-resolution results: I think these results are important, since this is the main motivating use case of COCO-GAN. We saw a drop in scores compared to PGGAN at the lower resolution. The question is that does that drop become larger when one goes to higher resolution---i.e., is a single generator with co-ordinate input still able to generate the whole image. As you say, fewer parameters is better, but only if they still give comparable performance.”
> > > > > > > > >
> > > > > > > > > We agree with the reviewer that an analysis toward real high-resolution image generation is important, and that is also our main motivation to run the CelebA-HQ experiment. However, note that we do not have sufficient computing power to perform extensive hyper-parameters tuning, and considering that GANs are highly sensitive to hyperparameters, it is hard to have a fair comparison in generation quality with PGGAN. In this work, we emphasize more on the contribution of demonstrating conditional coordinating is a possible solution and the generation quality is surprisingly better than expected. There may still be some unrecognized bottlenecks or design flaws not recognized by the authors, which requires more researchers and different opinions coming in for further study.
> > > > > > > > >
> > > > > > > > > “Question: for 1024x1024 full images, are you using 256x256 as the resolution of the micro patches ? What happens if you use the same hyper-parameters for the 128x128 dataset (i.e., 32x32 micro-patches) and use it to generate the 1024x1024 images ?”
> > > > > > > > >
> > > > > > > > > Yes, we use 256x256 resolution micro patches. We believe using 256x256 resolution means the same hyperparameters setting to 128x128 dataset as it is 1/4-sized on each of the edges of the full image.
> > > > > > > > > Using 32x32 micro patches to generate 1024x1024 images is similar to using 4x4 resolution micro patches to generate 128x128 CelebA images. This setting is difficult if not impossible to train a COCO-GAN as both the generator and the discriminator cannot observe useful information from the extremely-small partial views to learn accurate condition for coordinates.
> > > > > > > > >
> > > > > > > > > “This is again important in the context of the motivation of memory usage. Given a size of micro-patches that can be fit in memory, what factor larger images can one generate using COCO-GAN ? Is it limited to only 4x ?”
> > > > > > > > >
> > > > > > > > > No, it is not limited to 4x only, the task of panorama generation using 12x4 micro patches is an experimental proof. Smaller patch size may cause both the generator and the discriminator to fail to learn accurate condition for coordinates, since small patches may not form a valid partial view for the models to learn useful information. It forms a trade-off between patch size and generation quality as a hyperparameter.
> > > > > > > > >
> > > > > > > > > This is kind of similar to low-precision floating point training. Low precision training has many benefits (e.g., memory usage, computing power, and training time). But information loss of the low precision gradients causes many numerical problems, which many researchers are still working on reducing the side effects.

---

> > > > > > > > > ### Author Response · Authors · 2018-11-25
> > > > > > > > > **Rebuttal update**
> > > > > > > > >
> > > > > > > > > Thanks for all the reviewers’ effort in the paper review, we received lots of valuable suggestions. We accordingly revised our paper and create a meta-summary of the rebuttal. Please kindly check it out and leave some comments. We are more than willing to discuss more and further polish our paper in the remaining rebuttal period!
> > > > > > > > >
> > > > > > > > > Link to the meta-summary by the author:
> > > > > > > > > https://openreview.net/forum?id=r14Aas09Y7&noteId=ryxR2VUORX

---

> > > > > > > ### Author Response · Authors · 2018-11-14
> > > > > > > **Author's response [1/2]**
> > > > > > >
> > > > > > > “- So for the static graph generation, I think there may have been a mis-understanding. You don't need to generate a graph for all the crops (for training or generation). You would generate the graph for one crop (with the location of the crop being a placeholder at test time, or randomly generated for training). For generation, you would call the same graph multiple times with the same latent vector and different crop locations, and then just concatenate the results on the CPU---just like you're calling this paper's generator with different co-ordinates and concatenating them together. Training would be with one random crop at a time.”
> > > > > > >
> > > > > > >
> > > > > > > If we didn’t get the reviewer’s meaning wrong, take COCO-GAN as an example, it takes average 120 seconds to build the graph and average 7 seconds to run global initializer on a single GTX 1080 GPU. We believe it is hard to generate the graph on the fly for each iteration.
> > > > > > >
> > > > > > > As this is a little out of the scope, we hope the reviewers also agree that it would be better to prevent further discussions about the implementation detail of [M] here.
> > > > > > >
> > > > > > > “- About requiring conditional batch normalization (although, you could have different crops for different members in a batch---just like you have different co-ordinates in the current setting) and differences in parameters, yes, these are empirical questions---but that's why experiments would help to resolve this. It is possible that a cropping strategy may not work for GANs as it does for other tasks---or at least not as well as the proposed method, but again, experimental validation for this would be good.”
> > > > > > >
> > > > > > > We think the difference in parameters is a critical problem that should be taken into consideration when an algorithm is designed, and is not just an empirical issue. If the number of parameters is not a problem, then, indeed, conditional generation will not be a problem since we can train N generators for N classes. Weight sharing is still preferable in many cases if possible.

---

> > > > > ### Author Response · Authors · 2018-11-12
> > > > > **Thanks for the rapid response [1/2]**
> > > > >
> > > > >
> > > > > 1.
> > > > > “It's not clear why you'd need conditional batch-norm or why it would be an issue to implement in a static graph framework like Tensorflow. You could simply add a tf.slice (with fixed size crop for each layer, but random co-ordinates) after the output of every conv2d_transpose layer, and then do regular batch-normalization on that sliced output.
> > > > >
> > > > > Note that in the proposed method, batch-normalization is also being done on a smaller sized feature maps (corresponding to each crop)---it would be the same in this case, except that those smaller sized feature maps would come from cropping a larger map.”
> > > > >
> > > > > ** Conditional Batch Norm (CBN) **
> > > > > An empirical explanation is that the distribution of feature map crops are different in every spatial position, which requires different batch norm parameters to handle with (take CelebA as an example, the top side is mostly hair or hats, while the bottom side is mostly jaw or clothes). This is an empirical observation during COCO-GAN training, the model will result in extremely poor generation quality of without CBN. We expect [M] will have similar problems.
> > > > >
> > > > > We provide an anonymous image link showing how COCO-GAN might fail if it uses regular batch norm instead of CBN:
> > > > > https://imgur.com/ynzwIw4
> > > > >
> > > > >
> > > > > (We worry that the following section of our responses until the -*-*- sign would be too detailed and digressing from the main idea and contribution of our work; nevertheless, we hope the detailed discussions are helpful for illustrating some practical concerns about straightforward solutions. )
> > > > > ** Problems with static graph framework **
> > > > > We create a simple toy example in the anonymous link below to demonstrate the problem. The toy example considers our panorama generation setting, 64 batch size, 768x256 image size, and 64x64 patch size. To make things simple, we consider only the last convolutional layer that maps a 768x256x3 feature map to a 768x256x3 final output.
> > > > >
> > > > > An anonymous link to a toy example code:
> > > > > https://pastebin.com/8Eh0rcNz
> > > > >
> > > > > A - The first problem is building the graph. For static graph frameworks, everything should be done in the graph building stage. If the output of the generator is of size HxW, it will need to process almost HxW slices while graph building. In our toy example, the graph building is slow, which will become even slower as H and W grow. If with 10240x10240 input, it is estimated to take 1.5 days to build the graph on a single GTX 1080 GPU.
> > > > >
> > > > > B - The second problem is the training speed. In the toy example, forward pass without training takes 3.2 seconds per sample on a single GTX 1080 GPU. Our hypothesis about the low speed is that the complex graph (which introduces lots of extra Tensorflow operators, created 135,168 slice operators in the toy example only) and random selection makes the framework cannot take advantage of caching. Note that this result is for a single batch and a single layer only, which is already 2x slower than a full iteration of COCO-GAN training.
> > > > >
> > > > > C - The third problem is implementation. One will need to calculate the layer-wise receptive field for each position of the next layer, and meanwhile, needs to be aware of edge conditions. Furthermore, the feature map indexing changes for each modification on architecture, hyperparameters, and dataset. This is not ideal for software development.
> > > > >
> > > > > -*-*-*-*-*-*-*-*-*-*-*-*-*-*-*-*-*-*-*-*-*-*-*-*-*-*-*-*-*-*-*-*-*-*-*-*-*-*-*-*-*-*-*-*-*-
> > > > >
> > > > > Aside from the potential problems above, we may consider other cons (e.g. more parameters and training difficulty) and unknown implementation/training difficulties, [M] still requires researchers to further investigate before being used as a fair comparison target. Although [M] is straightforward at first glimpse, it is not simpler than or preferable to COCO-GAN from our point of view.
> > > > >
> > > > > If we consider the case that all methods share the same final generation quality, both [M] and training multiple generators introduce more parameters to be trained, while COCO-GAN suggests this can be done by conditional coordinate. It suffices that COCO-GAN provides a possible solution in a different dimension that the final generation quality is surprisingly good without post-processing. This is, to our knowledge, the first work observing this phenomenon and corroborating that COCO-GAN-like framework can generate images which are competitive to state-of-the-art methods.

---

### Author Response · Authors · 2018-11-25
**Rebuttal update**

Sincerely thanks for all the feedback from the reviewers. We summarize and respond to the reviewers’ suggestions and justifications in the following four aspects: writing, ablation study, providing more generated examples, and generating high-resolution images.

---

> ### Author Response · Authors · 2018-11-25
> **D. Generating high-resolution images**
>
> About validating generation quality directly on the high-resolution dataset, we decide to provide an anonymous link to a set of 1024x1024 samples generated by COCO-GAN. We run this experiment with our default hyperparameter configuration (this config was obtained from CelebA 64x64 experiment) on CelebA-HQ with 256x256 resolution patches.
>
> We are hesitant about providing these images as we are afraid if the community may have an unfair impression about  COCO-GAN’s generation quality. So far this experiment requires more hyperparameter tuning and the training may still not converge yet. We observe significant balancing problem in the G/D loss curve. Thus we suggest the results are taken as a reference only.
>
> Note that hyperparameter tuning for CelebA-HQ is very expensive and essential for GANs, but it is not affordable for us. For instance, PGGAN takes 2 weeks to train on CelebA-HQ, meanwhile, COCO-GAN should converge slower than PGGAN, since it does not adopt the “progressive growing” strategy.
>
> Anonymous link to CelebA-HQ 1024x1024 generation samples:
> https://www.dropbox.com/sh/5ly8xk22cqhxt76/AAAee1E2D8rIPAwFZtympnnta?dl=0

---

> ### Author Response · Authors · 2018-11-25
> **C. Providing more generated examples**
>
> As flagged by AnonReviewer1, we probably provide too few samples in the original paper, which causes the experimental result seems carefully curated rather than randomly sampled. We accordingly provide more samples at the following anonymous links. Note that the samples are provided in across epochs manner so that we can observe the seams disappears over epochs.
>
> CelebA 128x128 (micro patches 32x32, macro patches 64x64):
> https://www.dropbox.com/sh/tu2pf8qbnkklv53/AAB2YqnxDjAwdhRwOy2NtodLa?dl=0
>
> LSUN 256x256 (micro patches 64x64, macro patches 128x128):
> https://www.dropbox.com/sh/l7i9lxgdw8ez69v/AABA4ok5ldQM-B8FVgORpE5Oa?dl=0

---

> ### Author Response · Authors · 2018-11-25
> **B. Ablation study**
>
> We perform an ablation study and add a corresponding section (Section 3.3 and Appendix F) in the main paper based on all the reviewers’ suggestions. The ablation study is in two folds:
>
> 1. Comparison with a straightforward approach (refered to as [M] in the comment area and \mathcal{M} in the paper). The method [M] creates a full-sized generator, but trains and inferences with partial views by cropping corresponding feature maps during forward propagation. Despite the method [M] still implicitly uses the conditional coordinating strategy within the feature maps selection process, our experimental results in Table 1 suggest [M] and its variants (which are equipped with many COCO-GAN components at our best) cannot generate competitive results. The root cause of the poor results is unclear. Our hypothesis is the conditional batch normalization (CBN) in COCO-GAN is crucial for conditional coordinating. However, as the method [M] is not our main study target, we decide not to perform further analysis for improving the method [M].
>
> The FID curves through time are provided in the following anonymous link:
> https://imgur.com/wm6YZ1r
>
> Some generation samples through epochs are provided in the following anonymous link: https://www.dropbox.com/sh/pef4m1nz7wdo99i/AAB9dXVM0dMILU0Ojsw07Jl0a?dl=0
>
> Table 1: FID scores comparison between COCO-GAN and [M] method proposed by AnonReviewer3. All models are trained on CelebA dataset at 64x64 resolution. The results suggest COCO-GAN is more preferable in the same setting.
> -----------------------------------------------------------------------------------
> Model                                                                                         FID
> -----------------------------------------------------------------------------------
> COCO-GAN                                                                                4.99
> [M]                                                                                            72.82
> [M] + projection discriminator (100 epochs)                     90.87
> [M] + projection discriminator + macro discriminator    60.36
> -----------------------------------------------------------------------------------
>
>
>
> 2. We show an ablation study toward the trade-offs of each component in Table 2. The ablation study is conducted in the following five configurations:
>  - Continuous Sampling: Using the continuous uniform sampling strategy to sample spatial positions during training.
>  - Optimal Discriminator: The discriminator discriminates the full image, while the generator generates micro patches.
>  - Optimal Generator: The generator generates the full image, while the discriminator discriminates macro patches.
>  - Without Q Network: Removing the Q network which constructs the content consistency loss.
>  - Multiple Generators: Training an individual generator for each spatial position.
>
> We empirically found that the Q network that constructs the content consistency is not required if not considering the “Patch-Inspired Image Generation” application. Surprisingly, despite the convergence speed is different, the “Optimal Discriminator”, “COCO-GAN”, and “Optimal Generator” (ordered by convergence speed from fast to slow) can converge to similar final FID score if with sufficient training time. The difference in convergence speed is expected as “Optimal Discriminator” provides the generator with more accurate and global adversarial loss. In contrast, the “Optimal Generator” has relatively more parameters and layers to optimize, which causes the convergence speed slower than COCO-GAN. Lastly, “Multiple Generators” setting cannot converge well. Although it can also concatenate micro patches without obvious seams as COCO-GAN does, the full-image results often cannot agree and are not globally coherent. We provide the FID curve through time and the generated samples of the first 100 epochs in the following anonymous links:
>
> FID curve through time:
> https://imgur.com/BnVkouh
>
> Generated samples through epochs:
> https://www.dropbox.com/sh/y87ypswqslf9b5i/AAAmwXalLriX2ci5U7nkLxDZa?dl=0
>
> Table 2: FID scores comparison between COCO-GAN and its variants. All models are trained on CelebA dataset with the setting of 64x64 resolution, 16x16 micro patch, and 32x32 macro patch.
> -------------------------------------------------------------------------------
> Model                                                                          FID
> -------------------------------------------------------------------------------
> COCO-GAN (ours)                                                      4.99
> COCO-GAN (Continuous Sampling)                       6.13
> COCO-GAN (Optimal Discriminator)                      4.05
> COCO-GAN (Optimal Generator)                            6.12
> COCO-GAN (Remove Q Network)                           4.87
> Multiple Generators                                                  7.26
> -------------------------------------------------------------------------------

---

> ### Author Response · Authors · 2018-11-25
> **A. Writing**
>
> We are notified that our paper writing has some flaws, and it somewhat obscures the main targets and benefits of COCO-GAN. We accordingly have been revising the introduction and conclusion section of the paper to ensure the paper to be as clear as possible to the readers. We also fix some minor mistakes pointed out by the reviewers as follows:
>  - We create a section in the future work section mentioning and discussing “Synthesizing Images of Humans in Unseen Poses” by G. Balakrishan et al.
>  - We rewrite some inaccurate statements in Section 3.4 and add more citations.
>  - Figure 2 is replaced with other samples and referenced to more generated samples in Figure 12.
>
> Note that we made some other minor modifications to fit the 10-page limit.

---

### Meta-Review · Area_Chair1 · 2018-12-10
**metareview: interesting idea, experiments could be improved**

**Confidence:** 4
**Recommendation:** Reject

**Metareview:**

The paper introduces a GAN architecture for generating small patches of an image and subsequently combining them. Following the rebuttal and discussion, reviewers still rate the paper as marginally above or below the acceptance threshold.

In response to updates, AnonReviewer3 comments that "ablation experiments do make the paper stronger" but it "still lacks convincing experiments for its main motivating use case: generating outputs at a resolution that won't fit in memory within a single forward pass".

AnonReviewer2 points to the major shortcoming that "throughout the exposition it is never really clear why COCO-GAN is a good idea beyond the fact that it somehow works. I was missing a concrete use case where COCO-GAN performs much better."

Though authors provide additional experiments and reference high-resolution output during the discussion phase, they caution that these results are preliminary and could likely benefit from more time/work devoted to training.

On balance, the AC agrees with the reviewers that the paper contains some interesting ideas, but also believes that experimental validation simply needs more work, and as a result the paper does not meet the bar for acceptance.